# Review of Big Data and Processing Frameworks for Disaster Response Applications

**Silvino Pedro Cumbane *** and **Győző Gidófalvi**

Division of Geoinformatics, Department of Urban Planning and Environment,
KTH Royal Institute of Technology, Teknikringen 10A, SE-114 28 Stockholm, Sweden
*  Correspondence: silvino@kth.se; Tel.: +46-722774293

**Abstract:** Natural hazards result in devastating losses in human life, environmental assets and personal, and regional and national economies. The availability of different big data such as satellite imageries, Global Positioning System (GPS) traces, mobile Call Detail Records (CDRs), social media posts, etc., in conjunction with advances in data analytic techniques (e.g., data mining and big data processing, machine learning and deep learning) can facilitate the extraction of geospatial information that is critical for rapid and effective disaster response. However, disaster response systems development usually requires the integration of data from different sources (streaming data sources and data sources at rest) with different characteristics and types, which consequently have different processing needs. Deciding which processing framework to use for a specific big data to perform a given task is usually a challenge for researchers from the disaster management field. Therefore, this paper contributes in four aspects. Firstly, potential big data sources are described and characterized. Secondly, the big data processing frameworks are characterized and grouped based on the sources of data they handle. Then, a short description of each big data processing framework is provided and a comparison of processing frameworks in each group is carried out considering the main aspects such as computing cluster architecture, data flow, data processing model, fault-tolerance, scalability, latency, back-pressure mechanism, programming languages, and support for machine learning libraries, which are related to specific processing needs. Finally, a link between big data and processing frameworks is established, based on the processing provisioning for essential tasks in the response phase of disaster management.

**Keywords:** big data; processing frameworks; disaster response

## 1. Introduction

Disaster response is one of the most challenging phases of disaster management system since it addresses immediate threats presented by the disaster, including saving lives, meeting humanitarian needs (food, shelter, clothing, public health and safety), cleanup, damage assessment, task assignments and resource allocation. Recent study has shown that the amount of atmospheric greenhouse gas concentrations is increasing [1], and is unlikely to stabilize anytime soon [2]. Consequently, climate change is bound to continue and will cause severe natural disasters which support the idea that social, political and economic environment is as much a cause of disasters as the natural environment [3]. Severe natural disasters have been causing human suffering and deaths, massive infrastructures (e.g., buildings, roads, etc.) damages and negative economic impacts [4]. For instance, in 2015, earthquakes struck Nepal in April and May, killing just under 9000 people and injuring more than 22,000, with an estimated economic loss of around one-third of gross domestic product [5]. Recently, cyclone Idai hit at Beira, a low-lying port city in central Mozambique, causing widespread devastation

in southeastern Africa, which included 1006 deaths (603 in Mozambique, 344 in Zimbabwe, and 59 in Malawi), around 239,700 houses destroyed, and around 1.77 million acres of crops destroyed [6].

The main characteristics of natural hazards are complexity, unpredictability, and availability of limited resources in impacted areas [7]. The complexity is associated with the tasks that have to be carried out in each phase of disaster management, namely:

- mitigation (e.g., risk identification, analysis, public awareness and education, etc.);
- preparedness (e.g., emergency planning, training, early warning systems, etc.);
- response (e.g., searching, rescue operations, etc.); and
- recovery (e.g., rehabilitation and reconstruction).

The unpredictability of natural hazards is associated with the challenge of accurately predicting, for example, the severity of impacts of a disaster on people and infrastructures (for the response and recovery purposes), and the path of the hurricane (e.g., for evacuation before the disaster) [8]. The limited availability of resources brings the challenge on how to optimize the resources (human and material) allocation to minimize the impact of a disaster on people and their assets (during the mitigation and preparedness phases), as well as to reduce the number of deaths (during the response phase) [7].

Moreover, to minimize the impacts of disaster on people, an efficient and real-time disaster response system is needed. However, building such system is still challenging due to: (1) the variety of data that have to be integrated into the methods and models; and (2) the lack of reliable, scalable and interactive platform to increase the performance of disaster management systems [9]. In addition to that, choosing the "right" available big data framework for an application is also a big challenge [10].

Some systematic literature review studies on the application of big data for disaster management have been conducted. For instance, Selamat [11] discussed the usage of big data for disaster management, based on nine case studies, focusing on understanding the disaster management phases (prevention, preparedness, response, and recovery). From the analysis, the author concluded that big data are mainly used during the response phase. Arslan et al. [12], in addition to the disaster phases, considered the disaster type (floods, earthquakes, fire, hurricanes, and smog), data used (geoinformatics and sensor data, seismic intensity data, gas sensor data, Global Positioning System (GPS) traces, social media data, temperature data, population distribution data, video and socio-economic data), and key technologies used (Geographical Information Systems (GIS), cloud server, Mysql databases, NoSQL databases, Hadoop, ZigBee, Hive, Storm, and semantic networks) in each of twelve cases studies. Arslan et al. [12] found that there is a variety of big data available for each phase of disaster management and that linking different datasets with different kind of disasters is a big challenge. Recently, Yu et al. [13] presented a systematic literature review based on 149 selected articles from 101 journal articles. From this literature review, the authors established the link between disasters management phases, big data and disasters types.

However, there is no study that clearly shows *the link between big data used for disaster management and processing frameworks*. Therefore, this paper presents a systematic literature review on big data for disaster management, processing frameworks and establish a link between big data and processing frameworks, focusing on response phase of disaster management. This paper aims to support the disaster response authorities, mainly the data processors teams.

The remainder of the paper is organized as follows. Section 2 presents and discusses big data for disaster management. Section 3 presents and discusses the different big data processing frameworks and summarizes the main differences and similarities between the most popular approaches. Section 4 presents the link between big data and processing framework for disaster response. Finally, Section 5 concludes and presents future directions.

## 2. Big Data for Disaster Management

This section first describes the concept of disaster management, followed by a description of big data used for disaster management.

### 2.1. Disaster Management

Disaster management is defined as the coordination and integration of all activities necessary to build, sustain and improve the capabilities to prepare for, respond to, recover from, or militate against a disasters [14]. Essentially, it deals with the management of resources and information towards a disastrous event and is measured by how efficiently, effectively and seamlessly one coordinates these resources [15]. Therefore, the disaster management cycle comprises four phases divided into two groups, namely risk management (mitigation and preparedness) and crisis management (response and recovery) [12]. Mitigation phase is designed to reduce or eliminate risk, preparedness deals with planning the response to a disaster, response aims at maintaining or reestablishing public safety, and recovery consists on restoring the living conditions in the affected areas. To achieve the goals of each phase, specifics associated activities have to be carried out, for instance, mitigation consists in risk identification, analysis, and appraisal, and risk reduction by means of spatial planning, technical measures, and public awareness and education; **preparedness** focuses on emergency planning and training and installation and operation of monitoring, forecasting, and early warning systems; **response** consists of searching and rescue operations, and measures to provide for the basic humanitarian needs of the affected population; and **recovery** focuses on rapid damage assessment as well as rehabilitation and reconstruction [16]. Figure 1 presents disaster management phases and the activities associated with each phase.

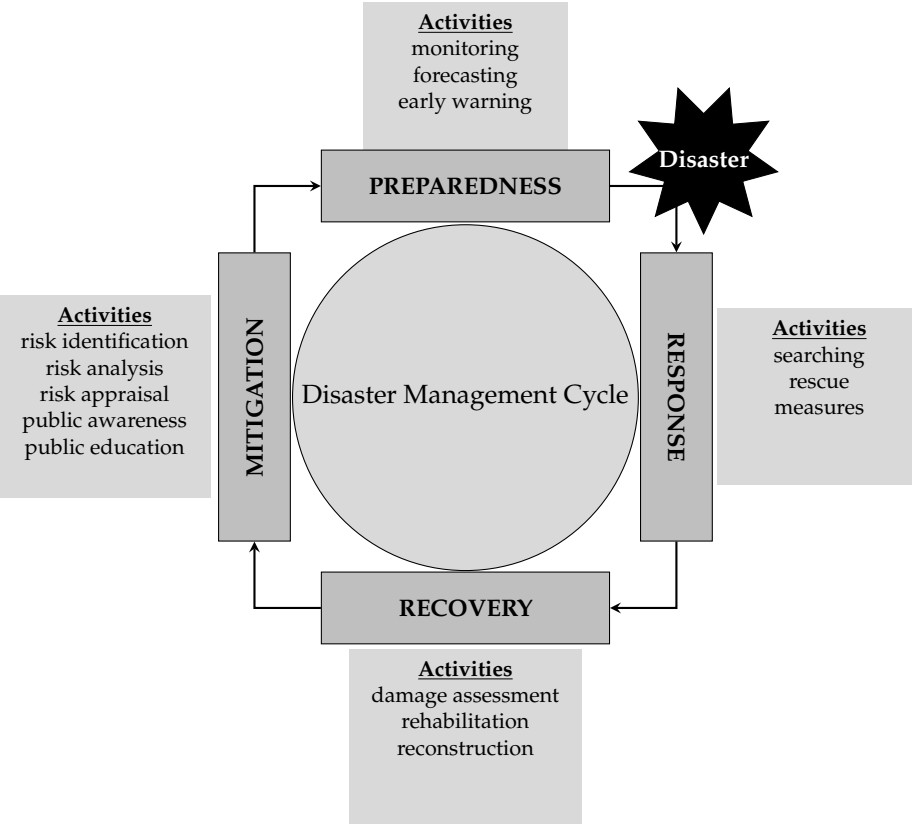

**Figure 1.** Disaster management cycle and associated activities from [16].

*2.2. Disaster Response*

Disaster response phase is mainly composed of two sub-phases, namely damage assessment and post-disaster coordination and response. Damage assessment consists of evaluating the affected areas to provide accurate information about the intensity of the damage which is crucial to guide the prioritization process of disaster responders. Post-disaster coordination and response consist of search and rescue operations. However, in many cases, some questions/challenges arise when it comes to addressing this phase: How can you quickly and effectively conduct the damage assessment, search, and rescue operations with limited resources? The answer to this question can be for instance using big data analytics since they provide possible solutions to understand the situations in disaster areas. However, these big data (e.g., satellite images, call detail records (CDR), crowdsourcing and social media) come from different sources (streaming data sources and data sources at rest), have different characteristics (spatial and non-spatial data), and different complexities. In addition, since some of these data are streams and have peaks (CDR) while others are even in distribution, the computation or processing framework that one wants to perform on these data sources are also different, e.g., some of them are continuous, likely window-based, simpler, stream computation while others might be acting on a dataset and are rather procedural and complex (e.g., task assignment/optimization). Moreover, some of the computations will nicely fit a Map-Reduce (MR) paradigm, while others (probably the optimization techniques) might need iteration/looping which MR is not inherently designed for. Figure 2 summarizes the questions/challenges that need to be answered to effectively address the disaster response.

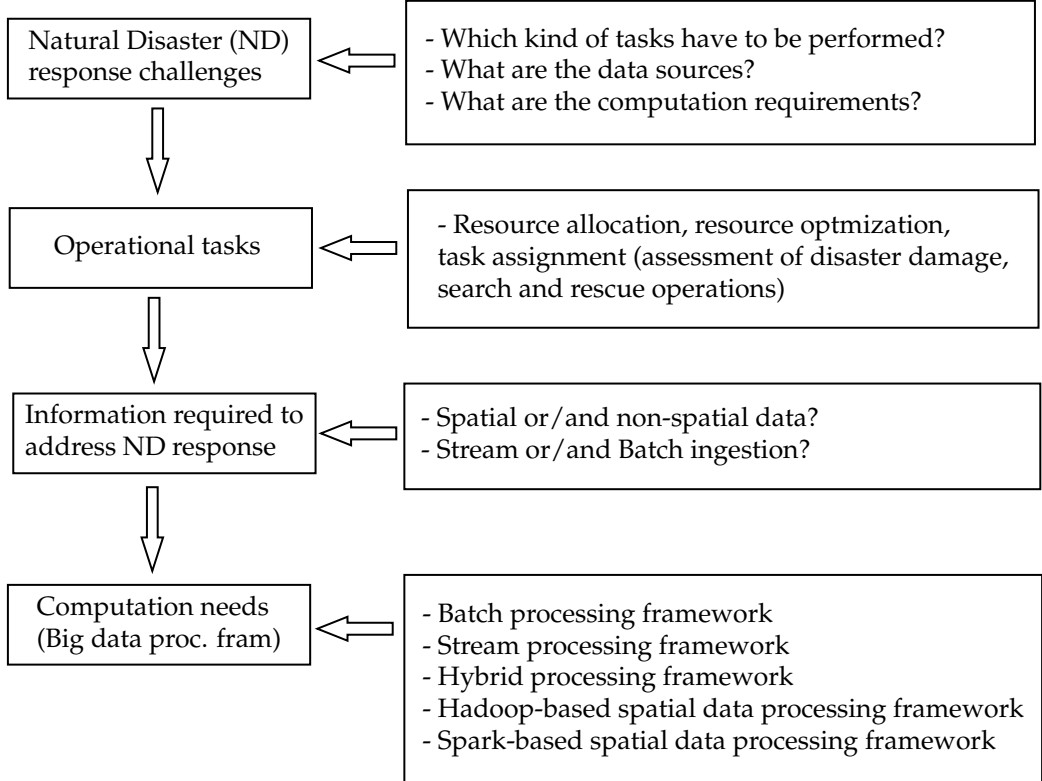

**Figure 2.** Disaster response questions or challenges.

To address these questions and challenges, description of big data used for disaster response to extract spatial and non-spatial information (e.g., affected areas, infrastructure damage assessment, people mobility, etc.) which are important for operational task assignment (e.g., resource allocation, saving lives, meeting humanitarian needs) and existing processing frameworks are presented below.

*2.3. Big Data and Disaster Response*

There is no standard definition of big data for disaster management [13] and consequently for disaster response. Generally, big data refers to large sets of complex data, both structured and unstructured which traditional processing techniques and/or algorithms are unable to operate on [17], i.e., the term "big data" is applied to datasets whose size is beyond the ability of commonly used software tools to capture, manage and process the data within a tolerable elapsed time [18]. The concept of big data is usually associated with three defining properties or dimensions, namely volume, velocity and variety well known as 3Vs ([19], Chapter 1). In addition to these three features, there are two characteristics: value, and veracity [20,21]. In other words, "big data represents the Information assets characterized by such a High Volume, Velocity and Variety to require specific Technology and Analytical Methods for its transformation into Value" [22]. **Volume** is related to overall data size, which goes from Gigabytes, Terabytes to Petabytes. **Velocity** refers to the pace of data being generated. **Variety** refers to different varieties of data formats being generated. **Value** highlights the economical outcome coming from data processing. **Veracity** refers to the quality and accuracy of data.

Satellite imagery, synthetic aperture radar (SAR), Wireless Sensor Web and Internet of Things (IoT), spatial data, crowdsourcing, social media records, and GPS traces and mobile Call Detail Record (CDRs), simulation, aerial imagery and video from Unnamed Aerial Vehicles (UAVs), airborne and terrestrial Light Detection and Ranging (LiDAR) have been reported as the major big data sources for disaster response [13], which are described next.

2.3.1. Satellite Imagery

Satellite imagery are images of the Earth collected by imaging satellite that can be used for post-disaster damage assessing through change detection [23], and disaster risk reduction, which includes landslide risk reduction [24], human settlement identification [25], and flood risk assessment [26]. These data are usually collected by passive sensors (sensors that measure reflected sunlight emitted from the sun). However, these sensors have limitations to capture the reflectance of Earth surface objects in cloud cover, rain conditions, and at nighttime. To overcome such limitations, active sensors such as the SAR can be effectively used to enlarge its observational capability during a natural disaster.

2.3.2. Wireless Sensor Web and Internet of Things (IoT)

Wireless Sensor Web is a group of dispersed and dedicated sensors that collect different kind of data such as temperature, humidity, wind, etc. These can be used to develop an early warning system for natural disasters [27] and to facilitate the communication between the affected population and rescue teams when traditional communication methods fail [28]. IoT-enabled devices such as *Grillo* (which means "cricket" in Spanish, is an earthquake alarming sensor network), *Brinco* (notification system for earthquake and tsunami), and *BRCK* (communication system under low connectivity areas) have been proposed as an alternative solution mainly in cases when there is a poor communication infrastructure in affected areas [29,30].

2.3.3. Crowdsourcing and Social Media

Crowdsourcing and social media are platforms contributed by the public. Therefore, while the public contributing in crowdsourcing platform are aware of it, contributors through social media are passive, which means that they are not aware of their contribution [31]. Moreover, while crowdsourcing platforms aim to improve disaster response and resource allocation based on real-time reports from affected people, social media are used to determine public sentiment and reaction to a disaster [13].

### 2.3.4. GPS Traces and Mobile Call Detail Record (CDRs)

GPS traces and mobile CDRs data have been used for different purposes in natural hazards management. Kafi and Gibril [32] reviewed applications of GPS trace data for different disasters, such as landslide monitoring (for monitoring of gradual changes in distance, height difference and coordinates of station within the area under study), tsunami monitoring (for building damage assessment), earthquake management (to monitor inter-seismic ground deformation and co-seismic displacement), forest fire (used to map fire perimeter and georeference the location and number of each damaged or destroyed structure), flood management (used together with remote sensing data and Geographic Information Systems for flood assessment, including the integration of inventory mapping, location of surface structures and roughness providing information on flow emplacement parameters). GPS trace data were also used to extract the behavior and mobility patterns after the occurrence of the Great East Japan Earthquake and Fukushima nuclear accident [33]. The authors found that people's mobility is impacted by intensity of disaster, the extent of damage, availability of government shelters, etc. In addition, they found that people's mobility and behavior after a large disaster sometimes is correlated with their mobility pattern during normal days.

Pastor-Escuredo et al. [34] investigated the viability of using mobile CDRs data combined with other sources of information to characterize the floods that occurred in Tabasco, Mexico in 2009. Using the variations in the number of active phones connected to each cell tower, the authors extracted the activity patterns in the most affected locations during and after the floods, which determined the signatures of the floods—in terms of both infrastructure impact assessment and population information awareness. In addition, mobile CDRs data were used to assess population displacement after 2015 Nepal Earthquake [35]. Using these data, it was possible to extract population mobility patterns after the earthquake and the patterns of return to affected areas, at a high level of detail.

### 2.3.5. Simulation

Modeling and simulation tools have been developed for disaster management with a focus on the response phase. For instance, Jain and McLean [36] proposed a framework for the integration of modeling, simulation, and visualization tools for emergency response. The goal of the proposed framework was to provide the whole picture of disaster to planners, trainers, and responders. Another study on simulation tools for disaster response was conducted by Massaguer et al. [37]. In this study, the authors proposed an augmented reality simulation environment for testing the effectiveness of IT solutions for disaster response using multi-agent simulation (modeling each person involved as an agent). In recent years, Dou et al. [38] proposed an agent-based framework for human rescue operations in landslide disaster events to evaluate the contingency plan. The proposed framework uses high-resolution remote sensing images, simulates a landslide environment based on a three-dimensional landslide geological model, and uses a multi-agent simulation approach to provide individuals' behavior simulation under dynamic disaster scenarios.

### 2.3.6. Unnamed Aerial Vehicles (UAVs), Drones and LiDAR

UAVs and drones provide high-resolution images and videos that can be applied to flood monitoring, fire detection, including intervention monitoring and also for post-fire monitoring, earthquakes rapid mapping and spread of hazardous materials and nuclear accidents [39]. Many sensors such as weather sensor, cameras, videos, infrared and ultraviolet sensor, among others, can be embedded in an UAV or drone and data-driven from these sensors can be used for real-time decision making about evacuation routes, damage assessment and transport logistics ([40], Chapter 1). In addition to that, LiDAR is a remote sensing based data acquisition method that can be used to generate detailed maps of topography and digital elevation models necessary for flood modeling and vulnerability and risks analysis. LiDAR can be applied for disaster monitoring, which includes flood prediction and assessment, monitoring of the growth of volcanoes and assistance in the prediction of

eruption, assessment of crustal elevation changes due to earthquakes, and monitoring of structural damage after earthquakes [41].

LiDAR technology provides the geospatial community with massive amounts of data for use in a variety of applications. As data collection continues, some challenges such as how to decrease data transmission, storage and processing requirements arise [18]. This description and challenges fit with other data sources such as satellite imagery, SAR, aerial imagery and video from UAVs, Wireless Sensor Web and IoT, simulation, crowdsourcing, social media records, and GPS traces and CDRs. In addition, LiDAR fits with big data description [18]. Therefore, satellite imagery, SAR, aerial imagery and video from UAVs, Wireless Sensor Web and IoT, simulation, crowdsourcing, social media records, and GPS traces and CDRs also fit with LiDAR description and, hence, with big data. Table 1 shows some examples of applications of big data for disaster response.

**Table 1.** Example of big data applications for disaster response [13].

| Disaster Management Sub-Phase | Data Source | Application Fields |
| --- | --- | --- |
| **Damage Assessment** | Satellite imagery, UAV and drones, social media, Wireless Sensor Web (WSW) and IoT, crowdsourcing | Earthquake, flood, typhoon, hurricane |
| **Post-Disaster Coordination and Response** | Social media, satellite imagery, WSW and IoT, UAV, crowdsourcing, simulation, LiDAR, GPS and CDRs, and combination of various data types | General natural disaster, flood and earthquake |

## 3. Big Data Processing Frameworks

This section is divided into two parts. Section 3.1 describes and compares the most popular big data processing frameworks. Section 3.2 presents and compares the processing frameworks used to manipulate big spatial data.

### 3.1. Popular Big Data Processing Frameworks

Many review papers that compare the most popular big data processing frameworks have been presented [42–45]. However, almost all compare processing frameworks without categorizing them according to the data sources that are designed to operate on them (batch, stream or hybrid). The study conducted by Gurusamy et al. [46] grouped the most popular big data processing frameworks into three clusters (batch-only, stream-only and hybrid) based on the state of the data they are designed to handle. However, this study does not explicitly present the aspects considered for the comparison of frameworks from the same group. Therefore, this paper uses the approach presented by Gurusamy et al. [46] to group the processing frameworks. Then, a summarized description of each processing framework is presented. Finally, a comparison of frameworks belonging to the same group is carried out considering the main aspects such as computing cluster architecture, data flow, data processing model, fault-tolerance, scalability, latency, back-pressure mechanism, programming languages, and support for machine learning libraries which are related to specific processing needs.

### 3.1.1. Batch Processing Frameworks

The term batch is often used to designate window of data i.e., a collection of data that have been grouped together within a specific time interval. Batch processing framework requires a set of data collected over time and all the data needed for the batch to be loaded to some type of storage, a database or file system to then be processed. Batch processing is often used when dealing with large volumes of data [47].

The most popular open-source batch processing framework today is Hadoop. It was proposed by Google and comes with MapReduce as its default engine [48]. Hadoop is composed of several

components that work together to process batch data, namely Hadoop Distributed File System (HDFS), Yet Another Resource Negotiator (YARN) and MapReduce [46].

HDFS is designed to store very large data sets reliably [49] and to ensure that the data remains available despite inevitable host failures [46]. It is also used as a source of data and to store intermediate processing results and make it available for the final computation. HDFS is composed of two architecture, namely master (*NameNode*) and slave (*DataNode*) [50]. The NameNode handles the responsibility of managing the namespace of the file system and governs the access by clients to files. The namespace records the creation, deletion and modification of files by users. NameNode maps data blocks to DataNodes and manages file system operations such as opening, closing and renaming of files and directories. Through the directions of NameNode, the DataNodes performs operations on blocks of data such as creation, deletion and replication.

YARN is a cluster which coordinates the components of Hadoop framework. The basic idea behind YARN is to split up the two major functionalities of the JobTracker, resource management and job scheduling into separate daemons [50].

MapReduce consists of two functions, namely *map* and *reduce*. The beauty of Hadoop MapReduce is that users usually only have to define the map and reduce functions. The framework takes care of everything else such as parallelization and failover. The Hadoop MapReduce framework utilizes a distributed file system to read and write its data. Typically, Hadoop MapReduce uses HDFS to store data and YARN to manage the resources and schedule the job [48], and its overall architecture can be found on [50].

### 3.1.2. Stream Processing Frameworks

Streaming processing deals with continuous data and is key to turning big data into fast data. It requires data to be fed into an analytics tool, often in micro-batches, and in real-time [47].

There is a considerable number of stream processing frameworks such as S4 [51] and more recent systems such as MillWheel [52], and Photon [53]. However, the most popular big data stream processing frameworks are Apache Storm and Apache Samza [46].

### Apache Storm

Storm is an open source framework for processing big data in real time built by Twitter. It is designed to be scalable, resilient, extensible, efficient and easy to administrate [54]. The main goal beyond the design is to avoid loss of a message due to node failures and to guarantee at-least-once processing [55].

The basic architecture of Storm consists of streams of tuples flowing through topologies which works by orchestrating DAGs (Directed Acyclic Graphs) where the vertices represent computation and the edges represent the data flow between the computation components. These topologies describe the various transformations or steps that will be taken on each incoming piece of data as it enters the system. According to Gurusamy et al. [46], the topologies are composed of streams, spouts, and bolts. **Streams** are conventional data streams, responsible for unbounded data that are continuously arriving at the system. **Spouts** are sources of data streams at the edge of the topology. These can be APIs, queues, etc. that produce data to be operated on. **Bolts** represent a processing steps that consumes streams, applies an operation to them, and outputs the result as a stream. Bolts are connected to each of the spouts and then connect to each other to arrange all of the necessary processing. At the end of the topology, the final bolt output may be used as an input for a connected system. To perform the processing task, Storm distributes bolts across multiple nodes to process the data in parallel.

Storm is based on two daemons called Nimbus (in master node) and a supervisor for each slave node. Nimbus keeps track of the progress of the worker nodes, supervises the slave nodes and assigns tasks to them. If it detects a node failure in the cluster, it reassigns the task to another node. Each supervisor controls the execution of its tasks (affected by the nimbus). It can stop or start spouts following the instructions of Nimbus. The coordination between supervisor nodes and the Nimbus

happens through the ZooKeeper. Each topology submitted to Storm cluster is divided into several tasks [56].

Topologies can be created using a high-level abstraction query model called **Trident**. This model provides high-level operators such as filters, joins, grouping, aggregations and functions [55]. In contrast to Storm, Trident API provides a stronger ordering guarantee, exactly-once processing semantics, works in micro-batches and introduces batch size as a parameter to increase throughput at the cost of latency. However, their topologies are not suitable for implementing iterative algorithms since they are directed acyclic graphs (DAGs) [57]. The architecture of Storm Trident can be found on [57].

Apache Samza

Apache Samza is an open source distributed processing framework developed by LinkedIn. This processing framework was created to solve various kinds of stream processing requirements such as efficient use of resources and at scale, handle failures gracefully, and scalability [58]. It provides at-least-once processing semantics and once-at-a-time processing model [57]. It uses Apache Kafka to provide fault tolerance, buffering and state storage [46], and Hadoop YARN for distributed resource allocation and scheduling [56]. The concept behind Kafka when dealing with data is based on five components, namely: **Producer**, **Topics**, **Consumer**, **Partitions**, and **Brokers** [46]. **Producer** write a topic to Kafka system. **Topic** is each stream of data entering the Kafka system that carry a key. The topic is read by a **Consumer** which is responsible for maintaining their own offset to be used in case a failure occurs. The incoming data/topic is divided into **Partitions** among the nodes based on key. Each node that makes up a Kafka cluster is called **Brokers**. Samza is based on three layers. The first one uses Apache Kafka to transmit the data flow. The second layer uses the YARN resource manager to handle the distributed execution of Samza processing and to manage CPU and memory usage across a multi-tenant cluster of machines. The processing capabilities are available in the third layer, which represents the Samza core and provides API for creating and running stream tasks in the cluster [56]. The architecture of Samza can be found on [57]. Differently from Storm, Samza does not need a back-pressure algorithm, it uses buffering data between processing steps, which makes intermediate results available to unrelated parties, for instance, other teams in the company [57].

Comparative Analysis between Storm, Storm Trident and Samza: Main Differences and Similarities

This section highlights the main differences and similarities between Storm, Trident and Samza in order to help in decision making about which streaming processing framework based on cluster architecture, data flow, data processing model, fault-tolerance, latency, scalability, back-pressure mechanism, programming languages, and support for machine learning libraries.

Computing Cluster Architecture

Storm and Storm Trident use Nimbus (Storm Master) for scheduling and monitoring tasks, Zookeeper for handling the coordination of tasks and JVM runs spouts and bolts as a task for each work. However, Samza lies on Kafka and Hadoop YARN to supervise one or more containers.

Data Flow

Storm uses a data pipeline called topology which consists of a directed graph where the nodes that ingest data and initiate the data flow are called spouts. Spouts emit tuples to downstream nodes called bolts. The bolts apply operations to tuples, and output is saved in external storage, and tuples are sent to further downstream bolts, and so on until the final output. However, Storm Trident Topologies (STT) are based on directed acyclic graphs (DAGs) since there is not support to cycles. Therefore, STT are less suitable for iterative algorithms. In addition, STT are not tuple-based data processing but micro-batch, which introduce a new parameter called batch size to increase the throughput. Samza lies on Kafka queuing system and messages entering the system is partitioned. Messages in the same partition are ordered, and there is no order guarantee for messages in different partitions. Differently

from what happens with Storm where the tuples are sent directly from one bolt to further downstream bolts, Samza output is written to Kafka. For further processing, Samza job may get the output of previous from Kafka as its input, and so on until the final output which is also written in Kafka.

Data Processing

Storm and Samza do provide at-least-once processing semantics. However, Storm does not give any guarantee on which order tuples will be processed while Samza offers sequential processing for messages in the same partition. However, Storm Trident provides exactly-once processing semantics and by default, batches are processed in sequence but it is possible to configure the parallel processing.

Fault-Tolerance

Recent Storm versions ($\geq$ 1.0.0) are fault-tolerant, i.e., provide state implementations that can recover from Supervisor's failure. If a bolt involved in processing does not acknowledge successful processing or if it explicitly signs a failure, Nimbus will take action to fix the problem. Storm Trident uses Storm's acknowledge feature to guarantee that each tuple is processed only once in the persistent state by maintaining additional information side state and by applying transactional updates [57]. Samza persists state in a local database and replicates state updates to Kafka. It uses periodical checking-points and reprocesses all data from a point ahead in case of failure registration.

Latency

In Storm, the topologies do not display end-to-end latency below 50 ms due to garbage collection and network latency impact ([59], Chapter 7). Even with the introduction of batch size, the periodic garbage collection that is triggered automatically may still have a negative impact of several milliseconds for small batches [60], therefore Storm Trident is a good choice for near real-time processing [46]. Even though persistence hop in single Kafka may only delay a message by milliseconds, complex analytics adds latency, which can be higher than compared to Storm implementations [57].

Scalability

To achieve scalability, Storm by default allows groupings that control data flow between nodes. The grouping consists of hash-partitioning or shuffling or arbitrary user-defined grouping of a stream of tuples by specific attribute value. Storm Trident allows user configuration for multiple batches processing in parallel. In Samza, scalability is achieved by running a Samza job in several parallel tasks, each of which consumes a separate Kafta partition and by avoiding the dynamic increase of the number of tasks during the runtime.

Back-pressure Mechanism

Storm version greater than 1.0.0 and Storm Trident have a back-pressure mechanism that controls the data flow and stops data ingestion when data are ingested faster than the processing capacity [57]. On the other hand, the use of Kafka and the possibility of buffering data between processing steps eliminate the need for a back-pressure algorithm for Samza.

Programming Languages

Storm and Storm Trident have a Java programming language based API that has adapters for numerous languages such as Python, Perl, and Ruby. However, Samza supports JVM languages, particularly Java.

Support for Machine Learning

Neither Storm nor Samza has a native machine learning library. However, these streaming processing frameworks support multi-language and user can use ML libraries available in the language

of your his/her choice. Both streaming processing frameworks are compatible with SAMOA API, which is a distributed streaming ML framework that contains a programming abstraction for distributed streaming ML algorithms [56]. However, Storm Trident has native ML library (Trident-ML) [61].

For better understanding, Table 2 summarizes the main differences and similarities among stream processing frameworks (Storm, Storm Trident and Samza).

**Table 2.** Comparison of stream processing frameworks.

| Framework | Storm | Trident | Samza |
|---|---|---|---|
| **Computing cluster architecture** | Nimbus | Nimbus | Hadoop YARN and Kafka |
| **Data Flow** | cyclic graph (stream - spout - bolt - bolt ... output) | directed acyclic graphs (DAGs) | Kafka - Kafka job - Kafka |
| **Data Processing Model** | at-least-once | exactly-once | at-least-once |
| **Fault-Tolerance** | Yes | Yes | Yes |
| **Latency** | several milliseconds | several milliseconds for small batches | several milliseconds |
| **Scalability** | Yes | User defined parallel processing | Yes |
| **Back-pressure mechanism** | Yes | Yes | No (uses buffering) |
| **Programming Languages** | Java API with adapters for Python, Ruby and Perl | Java API with adapters for Python, Ruby and Perl | Mostly uses Java |
| **Support for Machine Learning** | compatible with SAMOA API | Trident-ML | compatible with SAMOA API |

### 3.1.3. Hybrid Processing Frameworks

While some projects require one kind of processing frameworks, others may require both batch and stream workloads. In these cases, the usage of hybrid processing frameworks comes to action. The most popular hybrid processing frameworks are Apache Spark and Apache Flink.

Apache Spark

Apache Spark is hybrid processing framework built using similar principles of Hadoop's MapReduce engine with main goal of processing optimization by speeding up the batch processing workloads by full in-memory computation [46].

Spark interacts with storage layer only in the initial stage to load the data into memory and in the end of process to persist the final result. Unlike in Apache MapReduce, in Spark, all processing and intermediate results are, respectively, done and stored in-memory. Spark operation is based on distributed data structures called Resilient Distributed Datasets (RDDs). RDDs are fault-tolerant and automatically place tasks into partitions, maintaining the locality of persisted data. Beyond this, RDDs are versatile tools that allow programmers to persist intermediate results into memory or disk for reusability purposes. It also allows customizing the partitioning to optimize data placement. In case of data loss, each RDD is reconstructed based on the information on "lineage", which is responsible on tracking all the lazy operations performed by RDD [44].

Besides the Spark core, many libraries have been developed on top of the core to complement the basics functionalities of Spark. The most popular libraries are Machine Learning library (MLlib), Spark Streaming, Spark SQL, and Spark GraphX. **MLlib** is designed to simplify the ML pipeline in big data and its main functionalities include classification, regression, clustering, collaborative

filtering, optimization, and dimensionality reduction [62]. **Spark Streaming** allows the use of Spark's API to quickly process data, which can come from different data sources such as HDFS, Flume or Kafka in streaming environments by using mini-batches [44]. Spark Streaming divides input data streams into batches and stores them in Spark's memory. It then executes a streaming application by generating Spark jobs to process the batches [63]. **Spark SQL** is an Apache Spark module that integrates relational processing with Spark's functional programming API. It supports relational processing both within Spark programs (on native RDDs) and on external data sources using a programmer friendly API, provides high performance using established DBMS techniques, easily supports new data sources, including semi-structured data and external databases amenable to query federation, and enables extension with advanced analytics algorithms such as graph processing and machine learning [64]. **Spark GraphX** is an Apache Spark's module that combines the advantages of both data-parallel and graph-parallel systems by efficiently expressing graph computation within the Spark data-parallel framework. It consists on distributed graph representation to efficiently distribute graphs as tabular data-structures and takes the advantage of advances in data-flow systems to exploit in-memory computation and fault-tolerance [65]. The overall architecture of Apache Spark SQL can be found on [64].

Apache Flink

Apache Flink offers a hybrid data processing framework supported by his two main APIs namely DataStream and DataSet. It embraces data-stream processing as the unifying model for real-time analysis, continuous streams, and batch processing both in the programming model and in the execution engine [66]. It focuses on working with large data with very low data latency and high fault tolerance on distributed systems [44]. Flink programs can compute both early and approximate, as well as delayed and accurate, results in the same operation, obviating the need to combine different systems for the two use cases [66].

There are many libraries built on top of Flink processing framework, but the most popular are: FlinkML, Gelly, and Table API and SQL, FlinkCEP [44,67]. **FlinkML** is the machine learning library for Flink which aims to provide scalable machine learning algorithms and tools and primitives that help to design complex machine learning systems [68]. **Gelly** is the graph processing system in Flink which contains methods and utilities for the development of graph analysis applications. **Table API and SQL** is a SQL-like expression language for relational stream and batch processing enables the custom of SQL queries over the data. They are well embedded on both the DataStream and DataSets APIs and influence the usage of relational operators such as selection, aggregations, and joins. **FlinkCEP** is the complex event processing library which allows detecting complex events patterns in streams. The overall architecture of Apache Flink can be found on [66].

Comparative Analysis between Spark and Flink: Main Differences and Similarities

This section highlights the main differences and similarities between Spark and Flink in order to help in decision making about which hybrid processing framework to use based on cluster architecture, data flow, data processing model, fault-tolerance, latency, scalability, back-pressure mechanism, programming languages, and support for machine learning libraries.

Computing Cluster Architecture

Spark has high-level API that is composed by RDDs (Resilient Distributed Datasets) which are grouped in different API levels using Hadoop YARN or Apache Mesos. Flink offers web-based scheduling API to easily manage tasks and view the system. This processing framework can easily be integrated with YARN, HDFS, and Kafka for resources and tasks management.

Data Flow

Spark is based on micro-batch processing model which consists of a simple queue of RDDs called DStream. In this model, the incoming data are divided into small parts and processed one-at-time. To maintain the locality of persisted data, RDDS distributes operations on RDDs automatically into partitions. Moreover, RDDS give the possibility to programmers to persist intermediate results either into memory or disk for which can be used in the future in case of failure. This option customizes the partitioning to optimize data placement. However, in Flink, data are handled item-by-item as true stream [46]. In this processing framework, streams from any source enter in the system, where some functions are applied and new streams are generated. The output (new stream) flows out Flink system through Sinks that can be either database or another system. Moreover, Spark has ordering guarantee between batches while Flink provides within partition ordering guarantee.

Data Processing Model

While Flink is a native streaming processing framework with the possibility of batch processing, Spark was designed to work with static data with the possibility of real-time processing. However, both Spark and Flink provide exactly-once processing semantic. On the other hand, while Spark schedules its tasks using an acyclic graph approach in each iteration, Flink uses cyclic graph semantic to iteratively process the data. In addition, Flink uses one-at-a-time processing model.

Fault-Tolerance

Spark is fault-tolerance since it uses lineage to track lazy operations performed on each RDD. The tracked operations are used to reconstruct each RDD any moment in case of failure. Similar to Spark, Flink has a high fault-tolerance mechanism that recovers the state of data streaming in case of failure. This mechanism consists of generating snapshots of the distributed data stream and operator state which can be used by the system to fall back in case of failure.

Latency

Spark uses batch-stream processing model and notifies the scheduler at the end of each task. Invoking the scheduler at the end of each task adds overheads and results in decreased throughput and increased latency [69]. However, while operating at max throughput, Flink achieves a median latency of 26 milliseconds, and a 99th percentile latency of 51 ms, meaning that 99% of all latencies were below 51 ms [70].

Scalability

Spark parallelize the computation on user demand to achieve the scalability. However, Flink offers several APIs, which allows a user to consistently launch distributed computation in an easy manner [56]. This process consists on decomposing the job into a graph of operators, and the execution of each operator is physically decomposed into multiple parallel operator instances. For performance reasons, for iterative tasks, Flink carries out the computation on the node where the data are stored. In the case of a change in the data, it can do computation only on the changed data. As Flink, Spark also allows parallel processing to promote scalability in batch or stream big data.

Back-Pressure Mechanism

Spark and Flink offer back-pressure mechanism. In Spark, the back-pressure mechanism controls the receiving rate based on the current batch scheduling delays and processing times so that the system receives only as quickly as the system can process. Therefore, Back-pressure is a highly demanded feature that allows the ingestion rate to be set dynamically and automatically, based on previous micro-batch processing time [71]. Flink does not need a special mechanism for handling back-pressure, as data shipping in Flink doubles as a back-pressure mechanism. Therefore, Flink guarantees that

there are always enough buffers to make some progress, but the speed of this progress is determined by the user program and the amount of available memory [72].

Programming Languages

Spark is mostly written in Scala; however, it has API for Scala, Java and Python. Apache Flink is a distributed streaming dataflow engine written in Java and Scala.

Machine Learning

Spark and Flink provide Machine Learning (ML) facilities. Therefore, Spark has MLlib, which is a ML library that simplifies the ML pipeline in big data and allows performing classification, regression, clustering, and collaborative filtering, among others. On the other hand, Flink has a powerful ML library (FlinkML) designed to provide facilities for building complex ML systems.

For better understanding, Table 3 summarizes the main differences and similarities among hybrid processing frameworks (Spark and Flink).

**Table 3.** Comparison of hybrid processing frameworks.

| Framework | Spark | Flink |
|---|---|---|
| **Computing cluster architecture** | Hadoop YARN and Apache Mesos | Hadoop YARN and Kafka |
| **Data Flow** | simple queue of RDDs called DStream processed one-at-a-time using micro-batching cluster | stream $->$ system (operators) $->$ sinks |
| **Data Processing Model** | exactly-once | exactly-once |
| **Fault-Tolerance** | Yes (using lineage) | Yes (generating snapshots) |
| **Latency** | high | low |
| **Scalability** | Yes (on user demand) | Yes (parallelize the tasks that can be done in parallel) |
| **Back-pressure mechanism** | Yes | Yes |
| **Programming Languages** | API for Scala, Java and Python | Java and Scala |
| **Support for Machine Learning** | Yes (SparkMLlib) | Yes (FlinkML) |

*3.2. Big Spatial Data Processing Frameworks*

Geospatial information supports a wide range of government and community activities such as helping emergency authorities locate addresses and other important information so they can quickly respond [73]. This information is extracted through big spatial data and big spatial data processing frameworks. To analyze big spatial data to extract geospatial information, support for certain generic geospatial operations such as distance-based query, k-nearest neighbor (KNN) queries, filter-based queries (e.g., WithinDistance, Intersects, etc.), and others is needed. Therefore, big spatial data processing frameworks support these geospatial operations.

Big data processing frameworks originally deal with rednon-spatial data processing and analysis. Hadoop MapReduce based spatial data processing frameworks such as Parallel-Secondo [74], Hadoop-GIS [75], and SpatialHadoop [76] have been proposed to deal with spatial data processing and analysis. However, the most popular Hadoop-based spatial processing frameworks are Hadoop-GIS and SpatialHadoop. Similar to what happens with the non-spatial Hadoop MapReduce based processing framework, the spatial versions are also fault tolerant since for parallel processing the intermediate results are written in the disk. Therefore, there are six Spark based spatial processing frameworks namely Magellan [77], SpatialSpark [78], GeoMesa [79], GeoSpark [80], Simba [81], and

STARK [82]. The two most popular Spark-based spatial processing frameworks are GeoSpark and SpatialSpark.

### 3.2.1. Popular Hadoop-Based Spatial Processing Frameworks

Hadoop-GIS

Hadoop-GIS is a MapReduce based spatial data warehouse system for running large scale data partitioning and spatial queries [75]. Spatial queries include descriptive queries, spatial relationship based queries, distance-based queries, and spatial mining and statistics such as spatial clustering and spatial regression [83]. It utilizes SATO spatial partitioning (similar to KD-Tree) and local spatial indexing to achieve efficient query processing. However, it does not offer support of complex geometry types including convex/concave polygons, line string, multi-point, and multi-polygon. In addition, Hadoop-GIS only supports data up to two dimensions and two query types: box range queries and spatial joins over geometric objects with predicates such as within (within distance).

SpatialHadoop

To overcome Hadoop-GIS limitations, SpatialHadoop was proposed, which is a full-fledged MapReduce framework with native support for spatial data, spatial indexes and efficient spatial operations [84]. It extends MapReduce API with two new components, namely SpatialFileSplitter and SpatialRecordReader, for efficient and scalable spatial data processing. SpatialHadoop supports various geometry types, such as polygon, point, line string, multi-point, and others and multiple spatial partitioning techniques including uniform grids, R-Tree, Quad-Tree, KD-Tree, and Hilbert curves [85]. It also comes with several predefined spatial operations including box range queries, kNN queries and spatial joins over geometric objects using conditions such as within and intersect. In addition, SpatialHadoop supports different geometric objects, e.g., segments and polygons, and operations over them, e.g., generating convex hulls and skylines, which makes it a distributed geometric data analytics system over MapReduce.

Table 4 presents a comparison among the most popular Hadoop-based spatial big data processing frameworks. The comparison is based on the commonly used features such as spatial partitioning, spatial indexing, KNN query, distance query, distance join, filter (WithinDistance), etc. [78,86,87]. For instance, WithinDistance operator creates buffers using the buffer distance around the source features and returns all the features intersecting the buffer zones. For example, select infrastructures within 100 m of a river that will be flooded for pre-disaster evacuation.

**Table 4.** Comparison of Hadoop-based spatial big data processing frameworks.

| Feature | Hadoop-GIS | SpatialHadoop |
|---|---|---|
| **DataFrame API** | ✗ | ✗ |
| **In-memory processing** | ✗ | ✗ |
| **Spatial Partitioning** | SATO | Multiple |
| **Spatial Indexing** | R-Tree | R-/Quad-Tree |
| **KNN query** | ✓ | ✓ |
| **Query optimizer** | ✗ | ✗ |
| **Distance query** | ✓ | ✓ |
| **Distance join** | ✓ | ✓ |
| **Filter (Contains)** | ✓ | ✓ |
| **Filter (ContainedBy)** | ✓ | ✓ |
| **Filter (Intersects)** | ✓ | ✓ |
| **Filter (WithinDistance)** | ✓ | ✓ |

### 3.2.2. Popular Hadoop-based Spatial Processing Frameworks

SpatialSpark

SpatialSpark is built on top of Spark RDD to provide range query, spatial join, spatial filtering techniques and R-Tree index and R-Tree partitioning to speed up queries [78]. SpatialSpark is an in-memory big data system implementation, which is designed to support two spatial join operators, namely broadcast spatial join and partitioned spatial join [88]. Broadcast spatial join for joining one big dataset with another small dataset efficiently and partitioned spatial join for joining two big datasets. To achieve efficient joining, the right relation is indexed using an R-tre,ewhich is then made available to all worker nodes using Spark's broadcast variables. SpatialSpark provides Fixed Grid Partitioning, Binary Space Partitioning, and Sort Tile Partitioning with and without using an R-tree as the index to fetch the relations in case they do not fit into memory. It supports range queries with the predicates contains, within (containedBy), and withinDistance [78]. However, *k*-nearest neighbor (KNN) queries are not possible. In addition to that, there is no real API and many things still have to be done by hand by the user. Moreover, the result of a join returns only the matched pairs IDs, which requires additional joins afterward to retrieve the complete tuple in the application.

GeoSpark

GeoSpark is an in-memory cluster computing framework built on top of Spark for processing large-scale spatial data in a very fast manner compared to SpatialHadoop [86]. GeoSpark extends the RDDs concept and SparkSQL to support spatial data types, indexes, and geometrical operations at scale. GeoSpark provides a spatial RDD Application Programming Interface (API), which allows the Apache Spark programmers to easily develop their spatial analysis programs using operational (e.g., Java and Scala) and declarative (i.e., SQL) languages. It supports spatial data partitioning techniques such as uniform grid, R-tree, Quad-Tree, KDB-Tree, and KNN queries. GeoSpark comes with an optimizer which adaptively selects a proper join algorithm to strike a balance between the run time performance and memory/CPU utilization in the cluster [87]. Table 5 presents a comparison among the most popular Hadoop-based spatial big data processing frameworks.

**Table 5.** Comparison of Spark-based spatial big data processing frameworks.

| Feature | SpatialSpark | GeoSpark |
|---|---|---|
| **DataFrame API** | ✗ | ✓ |
| **In-memory processing** | ✓ | ✓ |
| **Spatial Partitioning** | Multiple | Multiple |
| **Spatial Indexing** | R-Tree | R-/Quad-Tree |
| **KNN query** | ✗ | ✓ |
| **Query optimizer** | ✗ | ✓ |
| **Distance query** | ✓ | ✓ |
| **Distance join** | ✓ | ✓ |
| **Filter (Contains)** | ✓ | ✓ |
| **Filter (ContainedBy)** | ✓ | ✗ |
| **Filter (Intersects)** | ✓ | ✓ |
| **Filter (WithinDistance)** | ✓ | ✗ |

## 4. Big Data and Processing Frameworks for Disaster Response

The big data concept raises new challenges such as how to analyze, archive, share, transfer and process large datasets across organizations [21]. To overcome such challenges, the big data paradigm provides data management tools and techniques for storage, processing and security the data. Therefore, the Big data paradigm can be divided into four main application areas: (i) big data methods, which focus on uncovering hidden trends and patterns from collection of data; (ii) big data storage, which deals with database management systems to store data; (iii) processing, which provides

data processing frameworks to do batch and stream processing; and (iv) representation, which provides visualizations based on the processed data [12]. This study focuses on the third application area (big data processing frameworks).

### 4.1. Link between Big Data and Processing Frameworks for Disaster Response

The disaster response phase consists of two sub-phases, namely: damage assessment and post-disaster coordination and response. Both sub-phases involve different big data (e.g., satellite imagery, UAV and drones, LiDAR, social media records, crowdsourcing, GPS traces, CDRs, etc.) that have to be integrated, analyzed, processed, and used by various teams to support a quick and effective decision making. Since these data are from different data sources, and have different characteristics and types (spatial and non-spatial), a challenge arises when it comes to deciding *which processing framework to use in order to extract useful information from the data* (e.g., people mobility estimation and prediction, affected areas estimation, infrastructures damage assessment, etc.). To address this challenge, it is important to know how the data will be injected (batch, streaming or hybrid ingestion) in the application program. Figure 3 summarizes the process of big data processing frameworks selection based on data type and data ingestion.

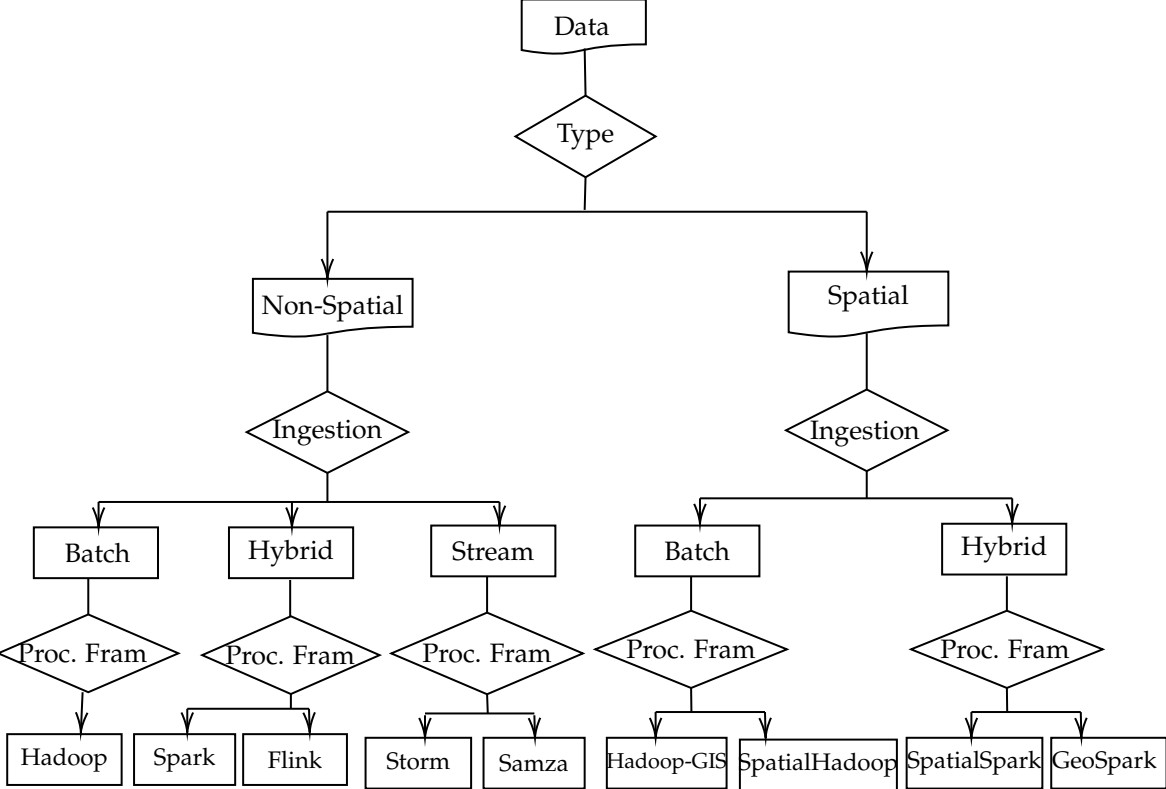

**Figure 3.** Big data processing framework selection flowchart.

However, in many cases, the selection process ends with more than one choice, namely: Storm and Samza in the case of streaming processing needs or Spark and Flink for hybrid processing needs for non-spatial data, and Hadoop-GIS and SpatialHadoop or SpatialSpark and GeoSpark for batch and hybrid spatial processing needs respectively. To select the most suitable processing framework for each case (spatial or non-spatial data) for a given application, detailed task requirements have to be considered. In the case of non-spatial data processing frameworks, the needs of support for machine learning, scalability, programming language, latency, etc., have to be evaluated. Tables 2 and 3 can be used to support that purpose. For the spatial data, the selection of the most suitable processing

framework will be based on the type of spatial operations that have to be performed (e.g., partitioning, distance query, filtering, etc.). Tables 4 and 5 can be used for that purpose.

*4.2. Use Case—Big Data and Processing Framework for Flooding Response*

This section presents a use case conceptual example of big data and processing frameworks for flooding response.

**Problem 1.** *Let us assume that there is a need to measure flooded areas (from Digital Elevation Model (DEM), land cover data, and satellite imagery), access the damage to structures (houses and roads) from crowdsourcing data and satellite imagery, estimate the precipitation from wireless sensors data, and there is a need to allocate the response team to aid the evacuation of population at risk based on a population density/mobility prediction model and potentially streaming location information in form of CDRs. Which processing frameworks should be used to implement all of this?*

Given that the data (mobile CDRs, satellite images, DEM, crowdsourcing and wireless sensors) are from different data sources (streaming data sources and data sources at rest), have different characteristics (spatial and non-spatial data), the information to be extracted from them (people mobility, affected areas, infrastructure damage, and precipitation) are complex, and they require both batch and stream processing frameworks, *Flink* and *Spark* are good candidates. Although both Spark and Flink are good candidates, the decision can be made based on user needs (latency, APIs, programming languages, etc.). For instance, from the latency point of view, Flink is better; however, Spark provides more APIs for the users. In addition to this, Spark offers spatial data analysis/operations (spatial partitioning, spatial index, KNN query, distance-based query and join, etc.) through SpatialSpark and GeoSpark processing frameworks, which are very important for the process of decision making. Therefore, Spark processing framework in conjunction with GeoSpark (which offers more spatial operations than SpatialSpark) can be suggested to perform these tasks. Figure 4 summarizes the data sources, data type, processing needs and output.

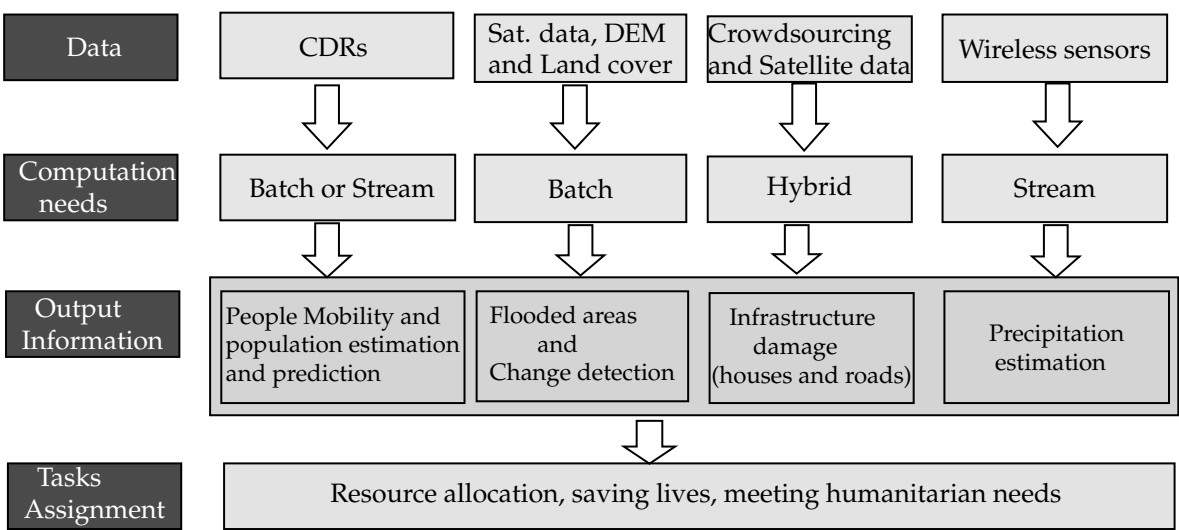

**Figure 4.** Big data and processing framework for disaster response.

## 5. Conclusions and Future Directions of Research

This paper presents an in depth analysis of existing big data processing frameworks. The paper is structured in three main parts that: describe the big data that have been used for disaster management, present the differences and similarities between most popular big data processing frameworks, and establish the link between big data and processing frameworks focusing on response phase of disaster management.

The first part of the paper identifies the main challenges of disaster response phase and describe the different big data that have been used to address some of the challenges. The second part group and describe different big data processing frameworks and presents the comparative analysis between processing frameworks belonging to the same group. It also presents and compares the popular big spatial data processing frameworks, which offer the spatial operations to efficiently deal with spatial data. The last part of this paper establishes a link between big data and processing frameworks focusing on the response phase of the disaster management cycle.

The use case shows that selecting the right processing frameworks to perform tasks using data from different data sources is challenging and require in-depth analysis of the characteristics and the complexity of the response system to be developed. In addition to that, the use of spatial operators needs to be analyzed for a better selection of spatial processing framework.

While a substantial amount of research has been invested in the application of big data in natural disaster management and link between big data and processing frameworks for disaster response (focusing on Flood), there is still a need of establishment of a link between big data, processing frameworks for different types of disasters (wildfire, flood, hurricane, earthquake, typhoon, landslide, volcano, etc.), and disaster management phases, to support the researchers and disaster management institutions.

**Author Contributions:** Silvino Pedro Cumbane and Győző Gidófalvi contributed in the study conception and design of literature review paper. Silvino Pedro Cumbane also contributed in acquisition of the reviewed papers, analysis of the literature review and the writing. In addition, Győző Gidófalvi provided comments for the revision of the paper.

**Funding:** This research received no external funding.

**Acknowledgments:** We thank the anonymous reviewers for their insightful comments, which significantly improved the paper.

**Conflicts of Interest:** The authors declare no conflict of interest.

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
