# Peer review of "Review of Big Data and Processing Frameworks for Disaster Response Applications"

_ijgi, doi:10.3390/ijgi8090387_

Round 1
Reviewer 1 Report
Although the authors thoroughly review in cloud-based big data processing frameworks, many information can be found in online documentation and computer books, especially Section 3. Hadoop, Apache Samza, Spark, Kafka, etc. are well documented in online resources. Many cloud-based big data analytics are designed for business applications such as consumer and customer behavioral analysis, credit fraud detection, etc., do not ideal for geospatial industry. There are a lot of challenges in geo big data processing with cloud. Nevertheless, some big data and cloud processing readers may interest.
Author Response
Dear Reviewer 1,
Please find the revised version of manuscript ID: ijgi-528869.
Reviewer 1 comments are valuable to improve the quality of this paper, and the paper has been revised according to Reviewer 1's comments. We believe that the paper is now of an acceptable standard after modification based on Reviewer 1's suggestions
The rest of this document is the details of our response to your comment:
Comment: Although the authors thoroughly review in cloud-based big data processing frameworks, many information can be found in online documentation and computer books, especially Section 3. Hadoop, Apache Samza, Spark, Kafka, etc. are well documented in online resources. Many cloud-based big data analytics are designed for business applications such as consumer and customer behavioral analysis, credit fraud detection, etc., do not ideal for geospatial industry. There are a lot of challenges in geo big data processing with cloud. Nevertheless, some big data and cloud processing readers may interest.
Response: Thank you very much for your comment. I agree with Reviewer 1 comment but the paper aims to give an overview of the general Big Data Processing Framework (BDPF) that is concise and accessible to the audience. Furthermore, the paper aims to place these frameworks in the context of disaster management for natural hazards in terms of data sources and their challenges (explained from the 4Vs big data perspective as well as from a geospatial perspective), the desired knowledge (analysis tasks), and knowledge generation methods (models etc.). Hopefully, by the removal of some of the generic big data processing architecture content, these application-specific aspects got more focus. Section 4.1 presents the application-specific content (the link between big data and processing frameworks for disaster response) which cannot be easily found in main BDPF references. In addition, section 4.2 presents a use case - big data and processing framework for flooding response. The use case explicitly presents the selection of the processing frameworks to process data from different data sources, with different characteristics to extract useful information to support flooding response authorities. Therefore, the extent of the coverage of the application-specific BDPF aspects, the data, and the knowledge will be overwhelmingly more than the generic BDPF aspects.
Reviewer 2 Report
The paper provides an interesting review of potential emerging technology available for disaster response applications.
In summary, the paper presented 4 objectives:
1. Big data sources
2. Big Data processing frameworks
3. Comparing big data processing frameworks
4. Link big data to processing framework sources
Positive Aspects of the paper:
The paper presents various big data types, and frameworks that can be applied in disaster management.
The paper compares frameworks and presents a flowchart to select the right big data processing framework.
Negative Aspects of paper:
I believe the paper meets some of the objectives but not all. The paper fails to present the frameworks in the context of disaster response and recovery. The paper does not flow in a way that is easily understandable. It reads more like a paper that is about big data processing frameworks. The links to disaster response are not clear.
The authors do not describe why big data should be applied in a disaster management context versus other data. Big data is not defined, in other words, GPS data in and of itself is not big data. A single satellite image is not big data, yet is still be used in disaster response. Further, the author describes big data gesospatial applications, but again does not situate the literature or big data processing frameworks in the disaster management context. The paper is geared toward selecting the proper framework, but it is developed based on the data, not the disaster. The flow of the big data processing framework should be based on the disaster type…and the kinds of data needed to respond to certain diasters. That is not the way the paper reads to me. In the conclusion, the author states there is still a need to establish the link between these frameworks and types of disasters.
The captions of the figures and tables are not very clear in terms of the purpose of the figures and tables. I am not clear on the relevance of some of the figures. For example: Figures 3-7 – Why is the architecture important to understanding linkages between big data and processing frameworks in the context of disaster response. In table 4, why are the features important in the context of disaster response?
There are many grammatical errors such as missing commas, run-ons or sentence tense mismatches. This takes away from the clarity of the paper, and disrupts the flow.
The paper does not describe potential stakeholders. Is this paper geared towards disaster responders? Or is it geared towards data processers?
I believe the author to start with the need for better management systems. Perhaps the paragraph at line 52 should be the first paragraph of the paper to set the stage for what the paper is about. That is the paragraph that defines the problem.
Acronyms should be spelled out or include a dictionary. i.e. GPS, GIS
Line 5: Why is geospatial important?
Paragraph 1 line 23 is a run-on sentence. It should be re-worded. The paragraph is also missing several commas.
Line 30: Natural disaster “usually” cause human suffering… Says who? I do not think “usually” is the right word.
Line 37: paragraph needs rewording
Line 45: No evidence to support claims
Line 47: Sentence needs rewording
Line 60: What are the “available features”?
Line 73: “the authors” which authors? Authors that you cited or authors of this actual review?
Line 75: you shouldn’t say “to the authors best knowledge” a review should be based on the existing evidence that you have reviewed.
Line 114: seems like this part should be moved to section 2.3
Line 126: What are the characteristics you are referring to? Characteristics of what? You mention characteristics and features, are they the same?
Line 158: You should define the acronyms throughout
Line 162: Lidar is used in damage assessment as well, but again, define Lidar and the other sources in terms of BIG data.
Line 173: sentence is incomplete and unclear
Line 192: what is population awareness and how does it relate to this review
Table 1: how is big data not applicable for damage assessment for all disaster management fields?
Line 215-216: presented where?
Line 217: “can handle” what?
Line 220: you say the study does not present the main factors…what are the main factors, your paper does not define these!
Line 236: Sentence is not clear
Line 238: What is a stack…this is not defined.
Figure 3: how is this relevant? What does word count process have to to with processing and/or big data and/or disaster management?
Line 254: lose = Loss!
Line 269 & Figure 4: Nimbus or numbus?
Line 271 what are spouts?
Figure 4-7: what is the relevance of the architecture of STORM etc?
How do the comparative analysis apply to the disaster management related data sources?
Section 3.1.2 Why do you have separate sections for comparisons?
Line 494: proposed where? & to who? And for what?
Table 4: good format for comparing the frameworks…but what is the relevance of the features? Situate this in the context of the disaster management
The conclusion does not really indicate what is best for what? The conclusion is very generic but does not offer any implications.
Author Response
Dear Reviewer 2,
Please find the revised version of manuscript ID: ijgi-528869.
Your comments are valuable to improve the quality of this paper, and the paper has been revised according to the comments of Reviewer 2. We believe that the paper is now of an acceptable standard after modification based on your suggestions
The rest of this document is the details of our response to your comments:
Comment 1: I believe the paper meets some of the objectives but not all. The paper fails to present the frameworks in the context of disaster response and recovery. The paper does not flow in a way that is easily understandable. It reads more like a paper that is about big data processing frameworks. The links to disaster response are not clear.
Response 1: Thank you very much for your comment. Reviewer 2 mentioned that the paper fails to present the frameworks in the context of disaster response and recover. The idea of this paper is to review the existing big data processing frameworks to help the researchers from disaster management field, particularly who deals with the response and recovery phase to select the processing frameworks for given big data source to perform certain tasks. To make it easier to read, in section 2.2. Disaster Response we added the following paragraph: “In order to address these questions and challenges, description of big data used for disaster response to extract spatial and non-spatial information (e.g., affected areas, infrastructure damage assessment, people mobility, etc.) which are important for operational task assignment (e.g., resource allocation, saving lives, meeting humanitarian needs) and existing processing frameworks are presented below.”
The link between Big Data and Processing Frameworks for Disaster Response is presented in section 4.1 and a use case on section 4.2 which describe with some degree of detail how to conduct the selection process of big data processing frameworks to perform some disaster response tasks.
Comment 2: The authors do not describe why big data should be applied in a disaster management context versus other data. Big data is not defined, in other words, GPS data in and of itself is not big data. A single satellite image is not big data, yet is still be used in disaster response. Further, the author describes big data geospatial applications but again does not situate the literature or big data processing frameworks in the disaster management context. The paper is geared toward selecting the proper framework, but it is developed based on the data, not the disaster. The flow of the big data processing framework should be based on the disaster type…and the kinds of data needed to respond to certain disasters. That is not the way the paper reads to me. In the conclusion, the author states there is still a need to establish the link between these frameworks and types of disasters.
Response 2: Thank you for your comment. The first paragraph of section 2.3 was re-worded in order to include the definition of big data as: “There is no standard definition of big data for disaster management [13] and consequently for disaster response. Generally, big data refers to large sets of complex data, both structured and unstructured which traditional processing techniques and/or algorithms are unable to operate on [17], i.e., the term "big data" is applied to datasets whose size is beyond the ability of commonly used software tools to capture, manage and process the data within a tolerable elapsed time [18]. The concept of big data is usually associated with three defining properties or dimensions, namely volume, velocity and variety well known as 3Vs [19, Chapter 1]. In addition to these three features, there are two characteristics: Value, and Veracity [20, 21]. In other words, “big data represents the Information assets characterized by such a High Volume, Velocity, and Variety to require specific Technology and Analytical Methods for its transformation into Value” [22]. Volume is related to overall data size, which goes from Gigabytes, Terabytes to Petabytes. Velocity refers to the pace of data being generated. Variety refers to different varieties of data formats being generated. Value highlights the economical outcome coming from data processing. Veracity refers to the quality and accuracy of the data.”
In addition, in order to address the comment “a single satellite image is not big data, yet is still be used in disaster response”, in section 2.3 was added the following paragraph: “LiDAR technology provides the geospatial community with a massive amount of data for use in a variety of applications. As data collection continue, some challenges such as how to decrease data transmission, storage and processing requirements arise [18]. This description and challenges fit with other data sources such as satellite imagery, SAR, aerial imagery and video from UAVs, Wireless Sensor Web and IoT, simulation, crowdsourcing, social media records, and GPS traces and CDRs. In addition, LiDAR fits with big data description [18]. Therefore, satellite imagery, SAR, aerial imagery and video from UAVs, Wireless Sensor Web and IoT, simulation, crowdsourcing, social media records, and GPS traces and CDRs also fits with LiDAR description and hence, with big data.”
Reviewer 2 mentioned that “the author describes big data geospatial applications, but again does not situate the literature or big data processing frameworks in the disaster management context”. I completely agree with Reviewer 2 but in this paper, we focused on establishing the link between big data and processing framework for disaster response application and as an example, in section 4.2. “Use Case - Big Data and Processing Framework for Flooding Response” we presented an example of how to select a “right” processing frameworks to perform given tasks of disaster response using different big data.
Reviewer 2 mentioned that “the paper is geared toward selecting the proper framework, but it is developed based on the data, not the disaster”. I agree with your comments, the paper focuses on big data and processing frameworks and presents a use case example (section 4.2) for disaster response (flooding response).
Reviewer 2 also mentioned that “the flow of the big data processing framework should be based on the disaster type…and the kinds of data needed to respond to certain disasters”. I completely agree with you. Therefore, in the conclusion we pointed out this aspect as the future direction of the review paper.
Comment 3: The captions of the figures and tables are not very clear in terms of the purpose of the figures and tables. I am not clear on the relevance of some of the figures. For example Figures 3-7 – Why is the architecture important to understanding linkages between big data and processing frameworks in the context of disaster response. In table 4, why are the features important in the context of disaster response?
Response 3: Thank you for your comment. The purpose of the architecture figures was to show how the frameworks are organized. However, based on your comment we decided to add sentences with references to authoritative sources about the architecture where the reader can obtain such information. The feature in table 4 helps to select the right spatial processing framework, for example, based on the type of spatial query we want to perform on the data (distance-based query, join based query, KNN query, etc.,) and/or type of data filter (contains, intersect, etc.).
Comment 4: There are many grammatical errors such as missing commas, run-ons or sentence tense mismatches. This takes away from the clarity of the paper, and disrupts the flow.
Response 4: Thank you for your comment. We went through the paper and corrected the grammatical error and the missing commas.
Comment 5: The paper does not describe potential stakeholders. Is this paper geared towards disaster responders? Or is it geared towards data processers?
Response 5: Thank you for your comment. To describe the potential stakeholders we added the following sentence in the introduction: “This paper aims to support the disaster response authorities mainly the data processors teams.”
Comment 6: I believe the author to start with the need for better management systems. Perhaps the paragraph at line 52 should be the first paragraph of the paper to set the stage for what the paper is about. That is the paragraph that defines the problem.
Response 6: Thank you very much for your comment. We reformulated the first paragraph as: “Disaster response is one of the most challenging phases of disaster management system since it addresses immediate threats presented by the disaster, including saving lives, meeting humanitarian needs (food, shelter, clothing, public health, and safety), cleanup, damage assessment, task assignments, and resource allocation. Despite that the human activities did not change much in the last years, the amount of atmospheric greenhouse gas concentrations is still increasing [1] and is unlikely to stabilize anytime soon [2]. Consequently, climate change is bound to continue and will cause severe natural hazards which support the idea that natural events are in the context of social, political and economic environments [3]. The sever natural hazards have been causing human suffering and deaths, massive infrastructures (e.g., buildings, roads, etc.) damages and negative economic impacts [4]. For instance, in 2015, earthquakes struck Nepal in April and May, killing just under 9,000 people and injuring more than 22,000, with an estimated economic loss of around one-third of gross domestic product [5]. Recently, cyclone Idai hit at Beira, a low-lying port city in central Mozambique, causing widespread devastation in southeastern Africa which included 1,006 deaths (603 in Mozambique, 344 in Zimbabwe and 59 in Malawi) and around 239,700 houses destroyed, around 1.77 million acres of crops destroyed[6].”
Comment 7: Acronyms should be spelled out or include a dictionary. i.e. GPS, GIS
Response 7: These aspects have been revised in the updated manuscript.
Comment 8: Line 5: Why is geospatial important?
Response 8: This part has been added as introduction 3.2. Big Spatial Data Processing Frameworks: “Geospatial information supports a wide range of government and community activities such as helping emergency authorities locate addresses and other important information so they can quickly respond [73]. This information is extracted through big spatial data and big spatial data processing frameworks. In order to analyze big spatial data to extract geospatial information, support for certain generic geospatial operations such as distance-based query, k-nearest neighbor (KNN) queries, filter-based queries (e.g., WithinDistance, Intersects, etc.), and others is needed. Therefore, big spatial data processing frameworks support these geospatial operations.”
Comment 9: Paragraph 1 line 23 is a run-on sentence. It should be re-worded. The paragraph is also missing several commas.
Response 9: Paragraph 1 has been re-worded as: “Disaster response is one of the most challenging phases of disaster management system since it addresses immediate threats presented by the disaster, including saving lives, meeting humanitarian needs (food, shelter, clothing, public health, and safety), cleanup, damage assessment, task assignments, and resource allocation. Despite that the human activities did not change much in the last years, the amount of atmospheric greenhouse gas concentrations is still increasing [1], and is unlikely to stabilize anytime soon [2]. Consequently, climate change is bound to continue and will cause severe natural hazards which support the idea that natural events are in the context of social, political and economic environments [3]. The sever natural hazards have been causing human suffering and deaths, massive infrastructures (e.g., buildings, roads, etc.) damages and negative economic impacts [4]. For instance, in 2015, earthquakes struck Nepal in April and May, killing just under 9,000 people and injuring more than 22,000, with an estimated economic loss of around one-third of gross domestic product [5]. Recently, cyclone Idai hit at Beira, a low-lying port city in central Mozambique, causing widespread devastation in southeastern Africa which included 1,006 deaths (603 in Mozambique, 344 in Zimbabwe and 59 in Malawi) and around 239,700 houses destroyed, around 1.77 million acres of crops destroyed[6].”
Comment 10: Line 30: Natural disaster “usually” cause human suffering… Says who? I do not think “usually” is the right word.
Response 10: These aspects have been revised in the updated manuscript. See paragraph 1 of the introduction.
Comment 11: Line 37: paragraph needs rewording
Response 11: This part has been revised in the updated manuscript as follow: “The main characteristics of natural hazards are complexity, unpredictability, and availability of limited resources in impacted areas [7]. The complexity is associated with the tasks that have to be carried out in each phase of disaster management namely:
· mitigation (e.g., risk identification, analysis, public awareness, and education, etc.);
· preparedness (e.g., emergency planning, training, early warning systems, etc.);
· response (e.g., searching, rescue operations, etc.); and
· recovery (e.g., rehabilitation and reconstruction).
Comment 12: Line 45: No evidence to support claims.
Response 12: Evidence added in the updated manuscript as: “The unpredictability of natural hazard is associated with the challenge of accurately predict, for example, the severity of impacts of disaster on people and infrastructures (for the response and recovery purposes), and the path of the hurricane (e.g., for evacuation before the disaster) [8]. The limited availability of resources brings the challenge on how to optimize the resources (human and material) allocation to minimize the impact of a disaster on people and their assets (during the mitigation and preparedness phases), as well as to reduce the number of deaths (during the response phase) [7].”
Comment 13: Line 47: Sentence needs rewording.
Response 13: This part has been revised in the updated manuscript.
Comment 14: Line 60: What are the “available features”?
Response 14: This part has been revised in the updated manuscript, and the sentence is now: “In addition to that, choosing the "right" available big data framework for an application is also a big challenge”.
Comment 15: Line 73: “the authors” which authors? Authors that you cited or authors of this actual review?
Response 15: These comments have been addressed.
Comment 16: Line 75: you shouldn’t say “to the authors’ best knowledge” a review should be based on the existing evidence that you have reviewed.
Response 16: This part has been revised in the updated manuscript as: “However, there is no study that clearly shows the link between big data used for disaster management and processing frameworks.”
Comment 17: Line 114: seems like this part should be moved to section 2.3.
Response 17: This comment has been addressed in the updated manuscript.
Comment 18: Line 126: What are the characteristics you are referring to? Characteristics of what? You mention characteristics and features, are they the same?
Response 18: This part has been revised in the updated manuscript as: “The concept of big data is usually associated with three defining properties or dimensions, namely volume, velocity and variety well known as 3Vs [18, Chapter 1].”
Comment 19: Line 158: You should define the acronyms throughout.
Response 19: These are not acronyms, are device names.
Comment 20: Line 162: Lidar is used in damage assessment as well, but again, define Lidar and the other sources in terms of BIG data.
Response 20: LiDAR technology provides the geospatial community with a massive amount of data for use in a variety of applications. As data collection continue, some challenges such as how to decrease data transmission, storage and processing requirements arise [18]. This description and challenges fit with other data sources such as satellite imagery, SAR, aerial imagery and video from UAVs, Wireless Sensor Web and IoT, simulation, crowdsourcing, social media records, and GPS traces and CDRs. In addition, LiDAR fits with big data description [18]. Therefore, satellite imagery, SAR, aerial imagery and video from UAVs, Wireless Sensor Web and IoT, simulation, crowdsourcing, social media records, and GPS traces and CDRs also fits with LiDAR description and hence, with big data.
Comment 21: Line 173: sentence is incomplete and unclear
Response 21: This comment has been addressed in the revised manuscript as: “Moreover, while crowdsourcing platforms aim to improve disaster response and resource allocation based on real-time reports from affected people, social media are used to determine public sentiment and reaction to a disaster [10].”
Comment 22: Line 192: what is population awareness and how does it relate to this review
Response 22: Thank you for your comment. The study presented by Pastor-Escuredo et al. [37] aims to extract signatures of the floods - both in terms of infrastructure impact assessment and population information awareness using Call Detail Records (CDRs). Therefore, population information awareness represents how the people were aware of the flood based on the activities pattern before and after the disaster. To make it much clear, we reworded this part in the revised manuscript as: “Using the variations in the number of active phones connected to each cell tower the authors extracted the activity patterns in the most affected locations during and after the floods which determine signatures of the floods - both in terms of infrastructure impact assessment and population information awareness.”
Comment 23: Table 1: how is big data not applicable for damage assessment for all disaster management fields?
Response 23: Table 1 was extracted in the study presented by Yu et al. [10] as shown in the table caption. In this study, the authors reviewed 149 selected articles to determine which data sources are used for each disaster management phase. Therefore, I think in their literature review they did not find relevant studies which show the application of some big data source for damage assessment.
Comment 24: Line 215-216: presented where?
Response 24: This part has been revised in the updated manuscript as: “Many review papers that compare the most popular big data processing frameworks have been presented [42–45].”
Comment 25: Line 217: “can handle” what?
Response 25: This comment has been addressed and the updated sentence is: “However, almost all compare processing frameworks without categorizing them according to data sources that are designed to operate on (batch, stream or hybrid).”
Comment 26: Line 220: you say the study does not present the main factors…what are the main factors, your paper does not define these!
Response 26: This part has been revised in the updated manuscript. The revised text is: “However, this study does not explicitly present the aspects considered for the comparison of frameworks from the same group. Therefore, in this paper will be used the approach presented by Gurusamyet al. [46] to group the processing frameworks. Then, a summarized description of each processing framework will be presented. Finally, a comparison of frameworks belonging to the same group will be carried out considering the main aspects such as computing cluster architecture, data flow, data processing model, fault-tolerance, scalability, latency, back-pressure mechanism, programming languages, and support for machine learning libraries which are related to specific processing needs.”
Comment 27: Line 236: Sentence is not clear
Response 27: The sentence has been revised in the updated manuscript as: “Through the directions of NameNode, the DataNodes performs operations on blocks of data such as creation, deletion, and replication.”
Comment 28: Line 238: What is a stack…this is not defined.
Response 28: In this sentence stack means framework. Therefore, to avoid confusion, we replaced the term stack by "framework" in the updated manuscript.
Comment 29: Figure 3: how is this relevant? What does word count process have to with processing and/or big data and/or disaster management?
Response 29: The idea was to provide an overview of how Hadoop processing framework works. Based on your comment we decided to exclude this Figure 3 from the updated manuscript and added a sentence with reference to authoritative sources about the architecture where the reader can obtain such information.
Comment 30: Line 254: lose = Loss!
Response 30: This part has been revised in the updated manuscript.
Comment 31: Line 269 & Figure 4: Nimbus or numbus?
Response31: This part has been revised in the updated manuscript.
Comment 32: Line 271 what are spouts?
Response 32: Spouts are sources of data streams at the edge of the topology. These can be APIs, queues, etc., that produce data to be operated on.
Comment 33: Figure 4-7: what is the relevance of the architecture of STORM etc?
Response 33: The idea was to provide an overview of the architecture of each processing framework so that the reader of this paper who may not be from the computer science field. However, we decided to exclude Figure 4-7. And have referred the reader to authoritative sources on the subject [57] – Figure 4 and 5, [64] – Figure 6, [66] – Figure 7.
Comment 34: How do the comparative analysis apply to the disaster management related data sources?
Response 34: The frameworks are grouped based on the data sources (stream data sources or/and data source at rest) that they operate on. Therefore, the comparative analysis is valid for disaster management related data sources or other kinds of big data sources.
Comment 35: Section 3.1.2 Why do you have separate sections for comparisons?
Response 35: The separate sections for comparisons are to make it easy to compare the frameworks from the same group (stream or hybrid).
Comment 36: Line 494: proposed where? & to who? And for what?
Response 36: This part has been revised in the updated manuscript as: “Therefore, there are 6 Spark based spatial processing frameworks namely: Magellan [68], SpatialSpark [69], GeoMesa [70], GeoSpark [71], Simba [72], and STARK [73].”
Comment 37: Table 4: good format for comparing the frameworks…but what is the relevance of the features? Situate this in the context of the disaster management.
Response 37: Thank you for your comment. The authors wrote the following paragraph: “Table 4 presents a comparison among the most popular spatial big data processing frameworks. The comparison is based on the commonly used features such as spatial portioning, spatial indexing, KNN query, distance query, distance join, filter (WithinDistance), etc [69,78,79]. For instance, WithinDistance operator creates buffers using the buffer distance around the source features and returns all the features intersecting the buffer zones. For example, select infrastructures (roads and buildings) within 100 meters of a river that will be flooded for pre-disaster evacuation.”
Comment 38: The conclusion does not really indicate what is best for what? The conclusion is very generic but does not offer any implications.
Response 38: Thank you for your comment. To meet your comment, the paper authors added the following paragraph: “The use case shows that selecting the right processing frameworks to perform tasks using data from different data sources is challenging and require in-depth analysis of the characteristics and the complexity of the response system to be developed. In addition to that, the use of spatial operators needs to be analyzed for a better selection of spatial processing framework.”
Reviewer 3 Report
This is a very interesting article and one which is increasingly relevant to the disaster management field. Below are a few points, which the authors should consider.
As a disaster studies scholar, I would suggest the authors consider removing the term "natural disasters" from the entire manuscript, and replace with either natural hazards, or simply disasters. The term natural disasters has long been excluded from disaster studies as it does not account for the complexity among hazards and social vulnerability and risk. See for instance the work of Susan L Cutter, see also :Blaikie P, Cannon T, Davis I, Wisner B. At risk: natural hazards, people’s vulnerability, and disasters. London: Routledge; 1994.
I do not see the need for such a long introduction paragraph on climate-induced extreme weather events. The authors don’t reference climate change anywhere else. If the authors are going to use this argument to imply that there will be an increase in the number of disasters owing to climate change, they could easily do so in a single sentence. What might be a better argument, however, would be to simply tie in the literature I reference above on increased social vulnerabilities to hazards (via exposure, social conditions, etc.).
On page 5, LiDAR could be integrated into the Sat imagery and/or UAV sections. Furthermore, the application fields in Table 1 are over simplified. Perhaps consider highlighting that those are merely some examples of hazard types where these data may be useful.
If the authors wish to engage with readers from less technical fields (including disaster studies) the article could do a better job of describing key technical terms early on to the readers, such as batch and stream processing. Even in the sections dedicated to these terms (3.1.1 and 3.1.2) clear definitions are not presented to the reader. It will also give the reader better context when the terms are related to disasters. For instance, this is done late in the paper on page 17 when the authors write "in order to address this challenge, it is important to know how the data will be injected (streaming or batch ingestion) in the application program." If could be used in the introduction of the paper, followed by simple definitions i.e. By streaming we mean, by batch we mean... etc. In the same way, it would be useful to briefly introduce Figure 8 (and perhaps even the flooding case example) in the beginning of the paper, along with brief definitions of the relevant processing framework elements (a single paragraph would suffice), to guide the reader and visually place the processing frameworks into the context of disasters.
Lastly, all of section 3 is written in domain specific language, which may make it difficult to understand the importance or relevance of the content for readers from other disciplines. I found that I was a bit lost among the comparisons with so many framework types and functions, as to what each function was and its level of relevance in disaster response. Perhaps this is unavoidable to some degree, but then consider if there is a way to better tie the conclusions from the comparisons in section 3 into what is now a rather short discussion/conclusion on disasters presented in sections 4 and 5.
Author Response
Dear Reviewer 3,
Please find the revised version of manuscript ID: ijgi-528869.
The comments are valuable to improve the quality of this paper, and the paper has been revised according to the comments of Reviewer 3. We believe that the paper is now of an acceptable standard after modification based on Reviewer 3’s suggestions.
The rest of this document is the details of our response to Reviewer 3’s comments:
Comment 1: As a disaster studies scholar, I would suggest the authors consider removing the term "natural disasters" from the entire manuscript and replace with either natural hazards, or simply disasters. The term natural disasters has long been excluded from disaster studies as it does not account for the complexity among hazards and social vulnerability and risk. See for instance the work of Susan L Cutter, see also: Blaikie P, Cannon T, Davis I, Wisner B. At risk: natural hazards, people’s vulnerability, and disasters. London: Routledge; 1994.
Response 1: Thank you very much for your comment. We have updated and changed the term “natural disasters” with “natural hazards”.
Comment 2: I do not see the need for such a long introduction paragraph on climate-induced extreme weather events. The authors don’t reference climate change anywhere else. If the authors are going to use this argument to imply that there will be an increase in the number of disasters owing to climate change, they could easily do so in a single sentence. What might be a better argument, however, would be to simply tie in the literature I reference above on increased social vulnerabilities to hazards (via exposure, social conditions, etc.).
Response 2: Thank you very much for your comment. We re-worded the introduction in general and the paragraph about the climate changes based on the reviewer 2 suggestion. The book suggested by reviewer was really interesting and is cited as in the first paragraph as [3]: “Disaster response is one of the most challenging phases of disaster management system since it addresses immediate threats presented by the disaster, including saving lives, meeting humanitarian needs (food, shelter, clothing, public health and safety), cleanup, damage assessment, task assignments, and resource allocation. A recent study has shown that the amount of atmospheric greenhouse gas concentrations is increasing [1], and is unlikely to stabilize anytime soon [2]. Consequently, climate change is bound to continue and will cause severe natural hazards which support the idea that natural events are in the context of social, political and economic environments [3]. Natural hazards have been causing human suffering and deaths, massive infrastructures (e.g., buildings, roads, etc.) damages and negative economic impacts [4]. For instance, in 2015, earthquakes struck Nepal in April and May, killing just under 9,000 people and injuring more than 22,000, with an estimated economic loss of around one-third of gross domestic product [5]. Recently, cyclone Idai hit at Beira, a low-lying port city in central Mozambique, causing widespread devastation in southeastern Africa which included 1,006 deaths (603 in Mozambique, 344 in Zimbabwe and 59 in Malawi) and around 239,700 houses destroyed, around 1.77 million acres of crops destroyed [6].”
Comment 3: On page 5, LiDAR could be integrated into the Sat imagery and/or UAV sections. Furthermore, the application fields in Table 1 are oversimplified. Perhaps consider highlighting that those are merely some examples of hazard types where these data may be useful.
Response 3: Thank you for your comment. We have integrated LiDAR into UAV and Drone section. In addition to that, we have highlighted that Table 1 shows some examples of applications of big data for disaster response.
Comment 4: If the authors wish to engage with readers from less technical fields (including disaster studies) the article could do a better job of describing key technical terms early on to the readers, such as batch and stream processing. Even in the sections dedicated to these terms (3.1.1 and 3.1.2) clear definitions are not presented to the reader. It will also give the reader a better context when the terms are related to disasters. For instance, this is done late in the paper on page 17 when the authors write "in order to address this challenge, it is important to know how the data will be injected (streaming or batch ingestion) in the application program." If could be used in the introduction of the paper, followed by simple definitions i.e. By streaming we mean, by batch we mean... etc. In the same way, it would be useful to briefly introduce Figure 8 (and perhaps even the flooding case example) in the beginning of the paper, along with brief definitions of the relevant processing framework elements (a single paragraph would suffice), to guide the reader and visually place the processing frameworks into the context of disasters.
Response 4: Thank you very much for your comment. We have added a short introduction paragraph for section 3.1.1 as: “Term batch is often used to designate window of data, i.e., a collection of data that have been grouped together usually within a specific time interval. Batch processing framework requires a set of data collected over time and all the data needed for the batch to be loaded to some type of storage, a database or file system to then be processed. Batch processing is often used when dealing with large volumes of data [44]”. We also added a short introduction paragraph for section 3.1.2 as: “Streaming processing deals with continuous data and is key to turning big data into fast data. It requires data to be fed into an analytics tool, often in micro-batches, and in real-time [44].”
Comment 5: Lastly, all of section 3 is written in the domain-specific language, which may make it difficult to understand the importance or relevance of the content for readers from other disciplines. I found that I was a bit lost among the comparisons with so many framework types and functions, as to what each function was and its level of relevance in disaster response. Perhaps this is unavoidable to some degree, but then consider if there is a way to better tie the conclusions from the comparisons in section 3 into what is now a rather short discussion/conclusion on disasters presented in sections 4 and 5.
Response 5: Thank you for your comment. In the conclusion (section 5) we added the following paragraph: "The use case shows that selecting the right processing frameworks to perform tasks using data from different data sources is challenging and require in-depth analysis of the characteristics and the complexity of the response system to be developed. In addition to that, the use of spatial operators needs to be analyzed for a better selection of spatial processing framework."
Round 2
Reviewer 2 Report
Line 1 losses not loses.
Line 26-28 The sentence is not clear. Natural hazards are not disasters. Thus severe natural “disasters” is a more appropriate word to use than hazards. The reference “Blaikie et al…” argues that the social, political and economic environment is as much a cause of disasters as the natural environment.
The author should be sure to indicate that they understand the difference between hazards and disasters. Hazards vs natural hazards, and disasters vs natural disasters are not interchangeable. i.e. All natural hazards are hazards, but all hazards are not natural.
Line 28 again refers to hazards instead of disasters. Hazards are the threat, disasters cause suffering.
Line 43 *hazards *predicting
Line 44 *of “a” disaster
Line 58 Reword: ie. …usage of big data for disaster management, based on nine case studies…focusing on understanding…
Line 59 “where the big data have been applied to” is not needed. The previous line already addresses that point
Line 66 Arslan et al… *found
Line 109 is missing commas
Line 157 *disasters
Line 218 *with massive
Line 219 *continues
Table 1 remove space between “Disaster” and “Management” in the table header
Lie 238 *Therefore, this paper will use…
Why are table 2 and 3 separate tables? Both compare processing frameworks
If table 2 is stream frameworks..that should be clear in the caption. The same for table 3, (hybrid) . Both captions simply say comparing …processing frameworks. You can have a table that compares the stream and hybrid. You do not have a table that compares batch. In table 4 you compare both batch and hybrid in the same table.
In table 3, differences are not really evident. What is the difference between “few” and “several”
Line 522 *non-spatial
Line 522 This paragraph is a bit confusing the way it is written. You should indicate after line 523, which are the popular frameworks…and likewise after line 527. Or…in line 529: you should separate the 4 based on the type of framework – Hadoop Map Reduce | Spark
Line 523 You name Hadoop Map Reduce frameworks:
1. Parallel-Secondo
2. Hadoop-GIS
3. SpatialHadoop
Line 527 You say therefore there are 6 Spark based, but then you only describe and compare 4 frameworks. In addition, only 2 of the 4 that you compare, come from the original 6.
1. Magellan
2. SpatialSpark
3. Geomesa
4. Geospark
5. Simba
6. STARK
You say the most popular are but these should be separated, as I indicated above.
1. Hadoop-GIS
2. SpatialHadoop
--
3. SpatialSpark
4. Geospark
In figure 3 you have a plus sign for the hybrid frameworks. I believe this is unclear in terms of the flow. How does it fit into the flow diagram…how can you reach a hybrid decision. (section 4.2 describes the need for hybrid data, but that comes much later in the text)
In figure 3 and 4 you use the terms Batch and Stream whereas the spatial frameworks that you described in the text are listed in Batch (Hadoop based) and Spark (hybrid based). This is a serious oversight that may lead to confusion.
This is also the case in section 4.1 where you indicate that the only type of ingestion is streaming or batch, and do not include hybrid.
If the figure and statements are correct, then it needs to be clear that spark based spatial frameworks are only stream processing and not hybrid.
Line 527 “In the beginning, there cannot be a single best option” why not? And when does the beginning become not the beginning anymore. There must be a way to make a decision especially in the use case that you describe in section 4.2 If I cannot make a decision at the beginning, how can I continue. I recommend that you reword the statements to assert that although both spark and flink are good candidates, you can make a decision based on user needs. APIs vs latency etc.
Figure 4 you have one box – [Batch/Stream] and another box saying [Batch and Stream] does “/” mean or? And does “and” mean hybrid?
Figure 4 also demonstrates that I need several frameworks to achieve the results for one use case. If I have many use cases, then the number will increase, how is this helpful to the agencies/users?
Line 637 I do not believe the paper is an in depth analysis of “research” it may be an in depth analysis of existing frameworks.
The introduction and conclusion claim that the paper establishes a link between big data and processing frameworks for disaster management, but 654 clearly contradicts that claim. “ there is no definite link between…” How can you establish a link then say there is no definite link.
Line 654 then follows by saying there needs to be a link established.
This is problematic. If you are claiming that there was no link established in previous research, than that needs to be clarified. In addition the conclusion should stress that a link was established. But I do not believe this is a strong claim.
Author Response
Dear Reviewer 2,
Please find the revised version of manuscript ID: ijgi-528869.
Your comments are valuable to improve the quality of this paper, and the paper has been revised according to the comments of Reviewer 2. The red-colored parts in the manuscript are newly added contents based on your valuable comments. We believe that the paper is now of an acceptable standard after modification based on your suggestions
The rest of this document is the details of our response to your comments:
Comment 1: Line 1 losses not loses.
Response 1: Thank you for your comment. This aspect has been revised in the updated manuscript.
Comment 2: Line 26-28 The sentence is not clear. Natural hazards are not disasters. Thus severe natural “disasters” is a more appropriate word to use than hazards. The reference “Blaikie et al…” argues that the social, political and economic environment is as much a cause of disasters as the natural environment.
Response 2: Thank you very much for your comment. These part has been revised in the updated manuscript and the sentence was re-worded as: “Consequently, climate change is bound to continue and will cause severe natural disasters which support the idea that social, political and economic environment is as much a cause of disasters as the natural environment [3].”
Comment 3: The author should be sure to indicate that they understand the difference between hazards and disasters. Hazards vs natural hazards and disasters vs natural disasters are not interchangeable. i.e. All natural hazards are hazards, but all hazards are not natural.
Response 3: Thank you very much for the comment. We went through the paper and we edited accordingly.
Comment 4: Line 28 again refers to hazards instead of disasters. Hazards are the threat, disasters cause suffering.
Response 4: This aspect has been addressed in the revised manuscript.
Comment 5: Line 43 *hazards *predicting
Response 5: These parts have been revised in the updated manuscript.
Comment 6: Line 44 *of “a” disaster
Response 6: These parts have been revised in the updated manuscript.
Comment 7: Line 58 Reword: ie. …usage of big data for disaster management, based on nine case studies…focusing on understanding…
Response 7: Thank you very much for the comment. This part has been revised in the updated manuscript.
Comment 8: Line 59 “where the big data have been applied to” is not needed. The previous line already addresses that point
Response 8: This part has been revised in the updated manuscript.
Comment 9: Line 66 Arslan et al… *found
Response 9: This part has been revised in the updated manuscript.
Comment 10: Line 109 is missing commas
Response 10: This comment has been addressed in the updated manuscript.
Comment 11: Line 157 *disasters
Response 11: This part has been revised in the updated manuscript.
Comment 12: Line 218 *with massive
Response 12: This part has been revised in the updated manuscript.
Comment 13: Line 219 *continues
Response 13: This comment has been addressed in the updated manuscript.
Comment 14: Table 1 remove space between “Disaster” and “Management” in the table header
Response 14: This part has been revised in the updated manuscript.
Comment 15: Lie 238 *Therefore, this paper will use…
Response 15: This comment has been addressed in the updated manuscript.
Comment 16: Why are table 2 and 3 separate tables? Both compare processing frameworks
Response 16: Thank you for the comment. We separated the table since they present a comparison of processing frameworks belonging to different groups (stream and hybrid).
Comment 17: If table 2 is stream frameworks..that should be clear in the caption. The same for table 3, (hybrid). Both captions simply say comparing …processing frameworks. You can have a table that compares the stream and hybrid. You do not have a table that compares batch. In table 4 you compare both batch and hybrid in the same table.
Response 17: Thank you for the comment. We edited the captions of table 2 and 3 in the updated manuscript. The idea of these tables is to compare the frameworks from the same group and since in the batch group, we have only one framework we did not create a table. In addition, we split table 4 into two (Table 4. Comparison of Hadoop-based Spatial Big Data Processing Frameworks and Table 5. Comparison of Spark-based Spatial Big Data Processing Frameworks).
Comment 18: In table 3, differences are not really evident. What is the difference between “few” and “several”
Response 18: This part has been revised in the updated manuscript.
Comment 19: Line 522 *non-spatial
Response 19: This comment has been addressed in the updated manuscript.
Comment 20: Line 522 This paragraph is a bit confusing the way it is written. You should indicate after line 523, which are the popular frameworks…and likewise after line 527. Or…in line 529: you should separate the 4 based on the type of framework – Hadoop Map Reduce | Spark
Response 20: This part has been revised in the updated manuscript as: “The 4 most popular Hadoop MapReduce- and Spark-based spatial processing frameworks are Hadoop-GIS and SpatialHadoop, and GeoSpark and SpatialSpark respectively.”
Comment 21: Line 523 You name Hadoop Map Reduce frameworks:
1. Parallel-Secondo
2. Hadoop-GIS
3. SpatialHadoop
Line 527 You say therefore there are 6 Spark-based, but then you only describe and compare 4 frameworks. In addition, only 2 of the 4 that you compare, come from the original 6.
1. Magellan
2. SpatialSpark
3. Geomesa
4. Geospark
5. Simba
6. STARK
You say the most popular are but these should be separated, as I indicated above.
1. Hadoop-GIS
2. SpatialHadoop
--
3. SpatialSpark
4. Geospark
Response 21: Thank you very much for the comment. To make it clear, we re-worded the paragraph as: “Big data processing frameworks originally deal with non-spatial data processing and analysis.Hadoop MapReduce based spatial data processing frameworks such as Parallel-Secondo [74],Hadoop-GIS [75], SpatialHadoop [76] have been proposed to deal with spatial data processing and analysis. However, the most popular Hadoop MapReduce-based spatial processing frameworks areHadoop-GIS and SpatialHadoop. Similarly with what happens with non-spatial Hadoop MapReduce-based processing framework, the spatial versions are also fault tolerance since for parallel processing the intermediate results are written in the disk. Therefore, there are 6 Spark-based spatial processing frameworks namely Magellan [77], SpatialSpark [78], GeoMesa [79], GeoSpark [80], Simba [81], andSTARK [82]. The 2 most popular Spark-based spatial processing frameworks are GeoSpark andSpatialSpark”.
Comment 22: In figure 3 you have a plus sign for the hybrid frameworks. I believe this is unclear in terms of the flow. How does it fit into the flow diagram…how can you reach a hybrid decision. (section 4.2 describes the need for hybrid data, but that comes much later in the text)
Response 22: Thank you for the comment. The Figure has been revised in the updated manuscript.
Comment 23: In figure 3 and 4 you use the terms Batch and Stream whereas the spatial frameworks that you described in the text are listed in Batch (Hadoop based) and Spark (hybrid based). This is a serious oversight that may lead to confusion.
Response 23: Thank you for the comment. Figure 3 and 4 have been revised in the updated manuscript to meet Reviewer 2’s comment.
Comment 24: This is also the case in section 4.1 where you indicate that the only type of ingestion is streaming or batch, and do not include hybrid.
If the figure and statements are correct, then it needs to be clear that spark based spatial frameworks are only stream processing and not hybrid.
Response 24: Thank you for the comment. This part has been revised in the updated manuscript.
Comment 25: Line 527 “In the beginning, there cannot be a single best option” why not? And when does the beginning become not the beginning anymore? There must be a way to make a decision especially in the use case that you describe in section 4.2 If I cannot make a decision at the beginning, how can I continue. I recommend that you reword the statements to assert that although both spark and Flink are good candidates, you can make a decision based on user needs. APIs vs latency etc.
Response 25: Thank you for the comment. This part has been revised as: “Although both Spark and Flink are good candidates, the decision can be made based on user needs (latency, APIs, programming languages, etc.). For instance, from the latency point of view, Flink is better however, Spark provides more APIs for the users.”
Comment 26: Figure 4 you have one box – [Batch/Stream] and another box saying [Batch and Stream] does “/” mean or? And does “and” mean hybrid?
Response 26: Yes, “/” mean or and “and” means hybrid. To avoid confusion, we have replaced “/” with “or” and Batch and Stream with “Hybrid” in the updated manuscript.
Comment 27: Figure 4 also demonstrates that I need several frameworks to achieve the results for one use case. If I have many use cases, then the number will increase, how is this helpful to the agencies/users?
Response 27: Yes, if we process the data separately, we will have to use several processing frameworks. However, in this use case, we wanted to show how to select the best framework/s able to process all the data. Therefore, Spark and Geospark were selected to perform all the data processing.
Comment 28: Line 637 I do not believe the paper is an in-depth analysis of “research” it may be an in-depth analysis of existing frameworks.
Response 28: Thank you for the comment. This part has been revised in the updated manuscript as: “This paper presents an in-depth analysis of existing big data processing frameworks.”
Comment 29: The introduction and conclusion claim that the paper establishes a link between big data and processing frameworks for disaster management, but 654 clearly contradicts that claim. “ there is no definite link between…” How can you establish a link then say there is no definite link.
Response 29: Thank you for the comment. The comment has been addressed in the updated manuscript. The paragraph has been re-worded as: "While a substantial amount of research has been invested in the application of big data in natural disaster management and link between big data and processing frameworks for disaster response (focusing on Flood), there is still a need of establishment of a link between big data, processing frameworks for different types of disasters (Wildfire, Flood, Hurricane, Earthquake, Typhoon, Landslide, Volcano, etc.), and disaster management phases, to support the researchers and disaster management institutions."
Comment 30: Line 654 then follows by saying there needs to be a link established.
This is problematic. If you are claiming that there was no link established in previous research than that needs to be clarified. In addition, the conclusion should stress that a link was established. But I do not believe this is a strong claim.
Response 30: Thank you for the comment. I agree with the comment but in this research, we only established a link between big data and processing frameworks focusing on the response phase of Flood. Therefore, we did not consider other types of disaster (Wildfire, Hurricane, Earthquake, Typhoon, Landslide, Volcano, etc.), and other phases of the disaster management cycle. That is why we claim that there is still a need of establishment of a link between big data, processing frameworks for different types of disasters (Wildfire, Flood, Hurricane, Earthquake, Typhoon, Landslide, Volcano, etc.), and disaster management phases, to support the researchers and disaster management institutions.